# Unveiling Invariances via Network Pruning

## Abstract

Invariance describes transformations that do not alter data's underlying semantics. Neural networks that preserve natural invariance capture good inductive biases and achieve superior performance. Hence, modern networks are handcrafted around well-known invariances (ex. translations). We propose a framework to learn novel network architectures that capture data-dependent invariances via pruning. Our learned architectures consistently outperform dense neural networks on both vision and tabular datasets in both efficiency and effectiveness. We demonstrate our framework on several neural networks across 3 vision and 40 tabular datasets.

## 1 Introduction

Preserving invariance is a key property in successful neural network architectures. Invariance occurs when the semantics of data remains unchanged under a set of transformations (Bronstein et al., 2017). For example, an image of a cat can be translated, rotated, and scaled, without altering its underlying contents. Neural network architectures that represent data passed through invariant transformations with the same representation inherit a good inductive bias (Neyshabur, 2020; 2017; Neyshabur et al., 2014) and achieve superior performance (Zhang et al., 2021; Arpit et al., 2017).

Convolutional Neural Networks (CNNs) are one such example. CNNs achieve translation invariance by operating on local patches of data and weight sharing. Hence, early CNNs outperform large multilayer perceptrons (MLP) in computer vision (LeCun et al., 2015; 1998). Recent computer vision works explore more general spatial invariances, such as rotation and scaling (Satorras et al., 2021; Deng et al., 2021; Delchevalerie et al., 2021; Sabour et al., 2017; Cohen & Welling, 2016; Jaderberg et al., 2015; Qi et al., 2017; Jaderberg et al., 2015; Xu et al., 2014). Other geometric deep learning works extend CNNs to non-Euclidean data by considering more data-specific invariances, such as permutation invariance (Wu et al., 2020; Kipf & Welling, 2016; Defferrard et al., 2016).

Designing invariant neural networks requires substantial human effort: both to determine the set of invariant transformations and to handcraft architectures that preserve said transformations. In addition to being labor-intensive, this approach has not yet succeeded for all data-types (Schäfl et al., 2022; Gorishniy et al., 2022; 2021; Huang et al., 2020). For example, designing neural architectures for tabular data is especially hard because the set of invariant tabular transformations is not clearly-defined. Thus, the state-of-the-art deep learning architecture on tabular data remains highly tuned MLPs (Kadra et al., 2021; Grinsztajn et al., 2022; Gorishniy et al., 2022).

Existing invariance learning methods operate at the data augmentation level (Immer et al., 2022; Quiroga et al., 2020; Benton et al., 2020; Cubuk et al., 2018), where a model is trained on sets of transformed samples rather than individual samples. This makes the network resiliant to invariant transformations at test time. Contrastive learning (CL) is a possible means of incorporating invariance (Dangovski et al., 2021), and has seen success across various tasks (Chen et al., 2021; Zhu et al., 2021; You et al., 2020b; Jaiswal et al., 2020; Baevski et al., 2020; Chen et al., 2020), including tabular learning (Bahri et al., 2021). While these approaches train model parameters to capture new data-dependent invariances, the model architecture itself still suffers from a weak inductive bias.

In contrast, existing network pruning works found shallow MLPs can automatically be compressed into sparse subnetworks with good inductive bias by pruning the MLP itself (Neyshabur, 2020). Combining pruning and invariance learning has largely been unsuccessful (Corti et al., 2022). Furthermore, pruning for invariance does not scale to deep MLPs, possibly due to issues in the lazy training regime (Tzen & Raginsky, 2020; Chizat et al., 2019) where performance improves yet

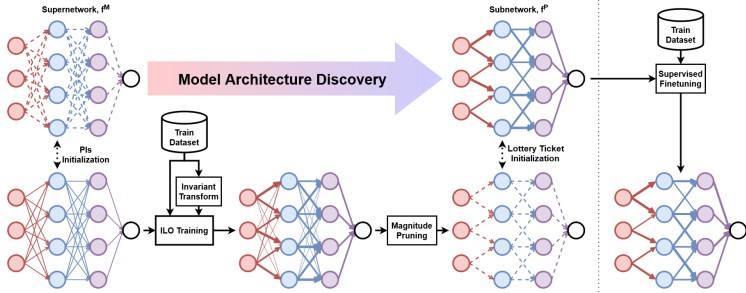

Figure 1: Overview for the IUNET Framework. The supernetwork, $f^P(\cdot; \theta_M)$, is initialized using PIs and trained on the ILO objective to obtain $\theta_M^{(T)}$. Magnitude-based pruning is used to get a new architecture $f^P = \mathcal{P}(\theta_M^{(T)})$. The new architecture, $f^P(\cdot; \theta_P)$, is initialized via lottery ticket reinitialization and finetuned with supervised maximum likelihood loss.

weights magnitudes stay near static over training. Combining invariance learning with network pruning remains an open question.

We propose **I**nvariance **U**nveiling Neural **Net**works, IUNET, a pruning framework that discovers invariance-preserving subnetworks from deep and dense supernetworks. We hypothesize pruning for invariance fails on deep networks due to the lazy training issue (Liu et al., 2023), where performance decouples from weight magnitudes. We address this with a proactive initialization scheme (PIs), which prevents important weights from being pruned through encouraging almost all weights to be near zero. To capture useful invariances, we propose a novel invariance learning objective (ILO), that successfully combines CL with network pruning by regularizing it with maximum likelihood.

To the best of our knowledge, we are the first to automatically design deep architectures that incorporate invariance using pruning. We summarize our contributions below:

- Designing architectures from scratch is difficult when desired invariances are hard to incorporate. We automatically discover an invariance-preserving subnetwork that outperforms an invariance-agnostic supernetwork on both vision and tabular data.

- Network pruning is used to compress models for mobile devices. Our approach consistently improves compression performance for existing vision and tabular models.

- Contrastive learning traditionally fails when combined with network pruning. We are the first to successfully combine contrastive learning with network pruning by regularizing it with our simple yet effective invariance learning objective.

- In the lazy training regime, performance improves drastically while weight magnitudes stay relatively constant, hence a weight's importance to downstream performance is decoupled from its magnitude. We provide an effective approach that encourages only important weights to have large magnitudes before the lazy training regime begins.

## 2 RELATED WORK

### 2.1 LEARNING INVARIANCES

Most invariant networks are handcrafted to capture specific spatial invariances (Dehmamy et al., 2021; Satorras et al., 2021; Deng et al., 2021; Qi et al., 2017; Vaswani et al., 2017; Cohen & Welling, 2016; Kipf & Welling, 2016; Jaderberg et al., 2015; LeCun et al., 1998). Learning invariance usually involves data augmentation followed by ensembling. (Immer et al., 2022; Quiroga et al., 2020; Lorraine et al., 2020; Benton et al., 2020; Cubuk et al., 2018). Some works use meta-learning to incorporate parameter sharing into a given architecture (Zhou et al., 2020; Kirsch et al., 2022). None of the aforementioned works generates architectures from scratch to improve the network's inductive bias. The closest work is $\beta$-LASSO (Neyshabur, 2020) which discovers shallow subnetworks with local connectivity through pruning for computer vision. Our work extends this idea to deeper networks and explores the tabular data setting.

| Dataset | $\text{MLP}_{\text{VIS}}$ | OMP $^{(\text{MLP}_{\text{VIS}})}$ | $\beta$-LASSO $^{(\text{MLP}_{\text{VIS}})}$ | IUNET $^{(\text{MLP}_{\text{VIS}})}$ |
|---------|------------|-------------|---------------|---------------|
| CIFAR10 | $59.266 \pm 0.050$ | $59.668 \pm 0.171$ | $59.349 \pm 0.174$ | $\mathbf{64.847 \pm 0.121}$ |
| CIFAR100 | $31.052 \pm 0.371$ | $31.962 \pm 0.113$ | $31.234 \pm 0.354$ | $\mathbf{32.760 \pm 0.288}$ |
| SVHN | $84.463 \pm 0.393$ | $85.626 \pm 0.026$ | $84.597 \pm 0.399$ | $\mathbf{89.357 \pm 0.156}$ |

| Dataset | RESNET | OMP $^{(\text{RESNET})}$ | $\beta$-LASSO $^{(\text{RESNET})}$ | IUNET $^{(\text{RESNET})}$ |
|---------|--------|-------------|---------------|---------------|
| CIFAR10 | $73.939 \pm 0.152$ | $75.419 \pm 0.290$ | $74.166 \pm 0.033$ | $\mathbf{83.729 \pm 0.153}$ |
| CIFAR100 | $42.794 \pm 0.133$ | $44.014 \pm 0.163$ | $42.830 \pm 0.412$ | $\mathbf{53.099 \pm 0.243}$ |
| SVHN | $90.235 \pm 0.127$ | $90.474 \pm 0.192$ | $90.025 \pm 0.201$ | $\mathbf{94.020 \pm 0.291}$ |

Table 1: Comparing different pruning approaches to improve the inductive bias of $\text{MLP}_{\text{VIS}}$ and RESNET on computer vision datasets. Notice, IUNET performs substantially better than existing pruning-based methods by discovering novel architectures that better capture the inductive bias. IUNET flexibly boosts performance of off-the-shelf models.

| Metric | $\text{MLP}_{\text{TAB}}$ | OMP | $\beta$-LASSO | IUNET | XGB | TABN | $\text{MLP}_{\text{TAB+C}}$ |
|--------|------------|-----|---------------|-------|-----|------|--------------|
| Num Top1 ↑ | 1 | 4 | 1 | 13 | 12 | 0 | **16** |
| Average Acc ↑ | 82.644 | 82.401 | 82.516 | **83.046** | 80.534 | 74.383 | 82.922 |
| Average Rank ↓ | 3.988 | 3.975 | 4.087 | 3.225 | 3.813 | 6.325 | **2.588** |

Table 2: We report the number of datasets out of 40 where each method was best, the average accuracy achieved by each method, and the average ranking of each method. OMP, $\beta$-LASSO, and IUNET all modify $\text{MLP}_{\text{TAB}}$. $\text{MLP}_{\text{TAB+C}}$ performed substantially more hyperparameter tuning than than IUNET. For full results, please refer to the Appendix.

## 2.2 NEURAL NETWORK PRUNING

Neural network pruning compresses large supernetworks without hurting performance (Frankle & Carbin, 2018; Louizos et al., 2017). A pinnacle work is the Lottery Ticket Hypothesis (LTH) (Frankle & Carbin, 2018; Liu et al., 2018b; Blalock et al., 2020), where pruned networks can retain unpruned peformance when reinitialized to the start of training and iteratively retrained. One-Shot Magnitude Pruning (OMP) studies how to prune the network only once (Blalock et al., 2020). The lazy training regime (Chizat et al., 2019) is a possible bottleneck for network pruning (Liu et al., 2023). Contrastive learning does not work with network pruning (Corti et al., 2022). Recent pruning policies improve efficiency by starting with a sparse network (Evci et al., 2020). or performing data-agnostic Zero-Shot Pruning (Hoang et al., 2023; Wang et al., 2020; Lee et al., 2019). Interestingly, subnetworks rarely outperform the original supernetwork, which has been dubbed the "Jackpot" problem (Ma et al., 2021). In contrast to existing works, we successfully combine OMP with contrastive learning, alleviate the lazy learning issue, and outperform the original supernetwork.

## 3 PROPOSED METHOD: IUNET

### 3.1 PROBLEM SETTING

We study the classification task with inputs, $x \in \mathcal{X}$, class labels, $y \in \mathcal{Y}$, and hidden representations, $h \in \mathcal{H}$. Our neural network architecture, $f(x; \theta) : \mathcal{X} \to \mathcal{Y}$ is composed of an encoder, $f_{\mathcal{E}}(\cdot; \theta) : \mathcal{X} \to \mathcal{H}$ and decoder, $f_{\mathcal{D}}(\cdot; \theta) : \mathcal{H} \to \mathcal{Y}$, where $\theta \in \Theta$ are the weights and $f = f_{\mathcal{E}} \circ f_{\mathcal{D}}$. During training, we denote the weights after $0 < t < T$ iterations of stochastic gradient descent as $\theta^{(t)}$.

First, we define our notion of invariance. Given a set of invariant transformations, $\mathcal{S}$, we wish to discover a neural network architecture $f^*(x; \theta)$, such that all invariant input transformations map to the same representation, shown in Equation 1. We highlight our task focuses on the discovery of novel architectures, $f^*(\cdot; \theta)$, not weights, $\theta$, because improved architectures capture better inductive bias, which ultimately improves downstream performance (Neyshabur, 2017).

$$f_{\mathcal{E}}^*(x; \theta) = f_{\mathcal{E}}^*(g(x); \theta), \forall g \in \mathcal{S}, \forall \theta \in \Theta. \tag{1}$$

| Dataset | $g(\cdot)$ | $\text{MLP}_{\text{VIS}}$ | $\text{IUNET}^{(\text{MLP}_{\text{VIS}})}$ |
|---|---|---|---|
| CIFAR10 | resize. | $44.096 \pm 0.434$ | $\textbf{97.349} \pm \textbf{4.590}$ |
| | horiz. | $80.485 \pm 0.504$ | $\textbf{99.413} \pm \textbf{1.016}$ |
| | color. | $56.075 \pm 0.433$ | $\textbf{98.233} \pm \textbf{3.060}$ |
| | graysc. | $81.932 \pm 0.233$ | $\textbf{99.077} \pm \textbf{1.598}$ |

| Dataset | $g(\cdot)$ | $\text{MLP}_{\text{TAB}}$ | $\text{IUNET}^{(\text{MLP}_{\text{TAB}})}$ |
|---|---|---|---|
| mfeat. | feat. | $46.093 \pm 1.353$ | $\textbf{51.649} \pm \textbf{4.282}$ |

Table 3: Comparing the consistency metric (%) of the untrained supernetwork, $\text{MLP}_{\text{VIS}}$ and $\text{MLP}_{\text{TAB}}$, against IUNET's pruned subnetwork under different invariant transforms, $g(\cdot)$. IUNET preserves invariances better.

## 3.2 FRAMEWORK

We accomplish this by first training a dense supernetwork, $f^M(\cdot; \theta_M)$, with enough representational capacity to capture the desired invariance properties, as shown in Equation 2. A natural choice for $f^M(\cdot; \theta_M)$ is a deep MLP, which is a universal approximator (Cybenko, 1989).

$$\exists \theta_M^* \in \Theta_M : f_{\mathcal{E}}^M(x; \theta_M^*) = f_{\mathcal{E}}^M(g(x); \theta_M^*), \forall g \in \mathcal{S}. \tag{2}$$

Next, we initialize the supernetwork's weights, $\theta_M^{(0)}$, using our Proactive Initialization Scheme, PIs, and train the supernetwork with our Invariance Learning Objective, ILO, to obtain $\theta_M^{(T)}$. We discuss both PIs's and ILO's details in following sections.

We construct our new untrained subnetwork, $f^P(\cdot; \theta_P^{(0)})$, from the trained supernetwork, $f^M(\cdot; \theta_M^{(T)})$, where the subnetwork contains a subset of the supernetwork's weights, $\theta_P^{(0)} \subset \theta_M^{(T)}$ and $|\theta_P^{(0)}| \ll |\theta_M^{(T)}|$, and is architecturally different from the supernetwork, $f^P(\cdot; \cdot) \neq f^M(\cdot; \cdot)$. For this step, we adopt standard One-shot Magnitude-based Pruning (OMP), where the smallest magnitude weights and their connections in the supernetwork architecture are dropped. We adopt OMP because of its success in neural network pruning (Frankle & Carbin, 2018; Blalock et al., 2020). We represent this step as an operator mapping supernetwork weights into subnetwork architectures $\mathcal{P} : \Theta_M \to \mathcal{F}_P$, where $\mathcal{F}_P$ denotes the space of subnetwork architectures.

$$f_{\mathcal{E}}^{P*}(x; \theta_{P*}) = f_{\mathcal{E}}^{P*}(g(x); \theta_{P*}), \forall g \in \mathcal{S}, \forall \theta_{P*} \in \Theta_{P*} \tag{3}$$

Finally, we re-initialize the subnetwork's weights, $\theta_P^{(0)}$, using the Lottery Ticket Re-initialization scheme (Frankle & Carbin, 2018) then finetune the subnetwork with maximum likelihood to obtain $\theta_P^{(T)}$. We hypothesize the trained subnetwork, $f^P(\cdot; \theta_P^{(T)})$, can outperform the trained original supernetwork, $f^M(\cdot; \theta_M^{(T)})$, if it preserves desired invariances and hence improves the inductive bias. The ideal subnetwork, $f^{P*}(\cdot; \theta_{P*})$, preserves invariances even without training, as shown in Equation 3.

We call this framework, including the ILO loss and PIs initialization, IUNET[1], as shown in Figure 1

### 3.2.1 INVARIANCE LEARNING OBJECTIVE: ILO

The goal of supernetwork training is to create a subnetwork, $f^P(\cdot; \theta_P^{(0)})$, within the supernetwork, $f^M(\cdot; \theta_M^{(T)})$, such that:

1. $\mathcal{P}(\theta_M^{(T)})$ achieves superior performance on the classification task after finetuning.

2. $\mathcal{P}(\theta_M^{(T)})$ captures desirable invariance properties as given by Equation 3.

3. $\theta_P^{(0)}$ has higher weight values than $\theta_M^{(T)} \setminus \theta_P^{(0)}$.

| Dataset | $\mathrm{MLP_{VIS}}$ | $\mathrm{IUNET}^{(\mathrm{MLP_{VIS}})}_{\mathrm{NO\text{-}PRUNE}}$ | $\mathrm{IUNET}^{(\mathrm{MLP_{VIS}})}_{\mathrm{NO\text{-}ILO}}$ | $\mathrm{IUNET}^{(\mathrm{MLP_{VIS}})}_{\mathrm{NO\text{-}PIS}}$ | $\mathrm{IUNET}^{(\mathrm{MLP_{VIS}})}$ |
|---|---|---|---|---|---|
| CIFAR10 | 59.266 | $54.622 \pm 0.378$ | $62.662 \pm 0.169$ | $60.875 \pm 0.292$ | $\mathbf{64.847 \pm 0.121}$ |
| CIFAR100 | 31.052 | $20.332 \pm 0.065$ | $32.242 \pm 0.321$ | $32.747 \pm 0.346$ | $\mathbf{32.760 \pm 0.288}$ |
| SVHN | 84.463 | $78.427 \pm 0.683$ | $88.870 \pm 0.139$ | $85.247 \pm 0.071$ | $\mathbf{89.357 \pm 0.156}$ |

| Dataset | $\mathrm{MLP_{TAB}}$ | $\mathrm{IUNET}^{(\mathrm{MLP_{TAB}})}_{\mathrm{NO\text{-}PRUNE}}$ | $\mathrm{IUNET}^{(\mathrm{MLP_{TAB}})}_{\mathrm{NO\text{-}ILO}}$ | $\mathrm{IUNET}^{(\mathrm{MLP_{TAB}})}_{\mathrm{NO\text{-}PIS}}$ | $\mathrm{IUNET}^{(\mathrm{MLP_{TAB}})}$ |
|---|---|---|---|---|---|
| arrhythmia | 67.086 | $56.780 \pm 6.406$ | $71.385 \pm 6.427$ | $\mathbf{78.675 \pm 7.078}$ | $74.138 \pm 2.769$ |
| mfeat. | 98.169 | $97.528 \pm 0.400$ | $\mathbf{98.471 \pm 0.344}$ | $98.339 \pm 0.203$ | $98.176 \pm 0.121$ |
| vehicle | 80.427 | $80.427 \pm 1.806$ | $81.411 \pm 0.386$ | $80.928 \pm 0.861$ | $\mathbf{81.805 \pm 2.065}$ |
| kc1 | 80.762 | $\mathbf{84.597 \pm 0.000}$ | $82.456 \pm 1.850$ | $\mathbf{84.597 \pm 0.000}$ | $\mathbf{84.597 \pm 0.000}$ |

Table 4: Ablation Study on vision and tabular datasets.

Because subnetworks pruned from randomly initialized weights, $\mathcal{P}(\theta_M^{(0)})$, are random, they include harmful inductive biases that hinders training. Thus, we optimize the trained supernetwork, $f^M(\cdot; \theta_M^{(T)})$, on goals (1) and (2) as a surrogate training objective. Goal (3) is handled by PIs, described in the next section.

To achieve (1), we maximize the log likelihood of training data. To achieve (2), we minimize the distance between representations of inputs under invariant perturbations, stated in Equation 5. Intuitively, achieving (2) entails optimizing the supernetwork in metric space, which we find is equivalent to Supervised Contrastive Learning (SCL) as state in Theorem 1.[2]

**Theorem 1** *Minimizing the distance between representations of inputs under a set of invariant perturbations, Equation 4, is equivalent to minimizing the supervised contrastive learning objective, Equation 5, where $f_{\mathcal{E}}^M : \mathbb{R} \to \mathbb{R}^d$ is a supernetwork, $\psi^{(cos)} : \mathbb{R}^d \times \mathbb{R}^d \to \mathbb{R}$ is cosine similarity, $\phi : \mathbb{R}^d \times \mathbb{R}^d \to \mathbb{R}$ is a distance metric, and $g : \mathcal{X} \to \mathcal{X}$ is a desired invariance function from $\mathcal{S}$.*

$$\theta_M^* = \underset{\theta_M}{argmax} \ \underset{\substack{x_i, x_j \sim \mathcal{X} \\ g \sim \mathcal{S}}}{\mathbb{E}} \left[ \frac{\phi(f_{\mathcal{E}}^M(x_i; \theta_M), f_{\mathcal{E}}^M(x_j; \theta_M))}{\phi(f_{\mathcal{E}}^M(x_i; \theta_M), f_{\mathcal{E}}^M(g(x_i); \theta_M))} \right] \tag{4}$$

$$= \underset{\theta_M}{argmin} \ \underset{\substack{x, y \sim D_{tr} \\ g \sim \mathcal{S}}}{\mathbb{E}} \left[ -log \left( \frac{exp\left( \psi^{(cos)}\left( f_{\mathcal{E}}^M(x; \theta_M), f_{\mathcal{E}}^M(g(x); \theta_M) \right) \right)}{\sum\limits_{\substack{x', y' \sim D_{tr} \\ y' \neq y}} exp\left( \psi^{(cos)}\left( f_{\mathcal{E}}^M(x; \theta_M), f_{\mathcal{E}}^M(g(x'); \theta_M) \right) \right)} \right) \right] \tag{5}$$

Explicitly optimizing both (1) and (2) is necessary for IUNET. Because maximum likelihood on its own does not consider desired invariance properties, pruning will not improve the inductive bias of supernetworks trained solely to optimize (1). For this reason, performance degradation is commonly observed amongst almost all existing pruning algorithms (Hooker et al., 2019; Blalock et al., 2020; Ma et al., 2021). Because pruning already causes the supernetwork to "selectively forget" training samples disproportionately (Hooker et al., 2019) and supernetworks trained solely with contrastive learning amplifies this effect (Corti et al., 2022), pruning will not improve performance of supernetworks trained solely to optimize (2). One reason why contrastive learning amplifies "selective forgetting" is because models overfit contrastive objectives (Zhang et al., 2020; Pasad et al., 2021).

By optimizing both (1) and (2), IUNET uses pruning to enhance the supernetwork by encoding helpful inductive biases into the pruned subnetwork while avoiding overfitting of the contrastive objective. The Invariance Learning Objective (ILO) is shown in Equation 7, where $\mathcal{L}_{NCE}$ is the contrastive loss defined in Equation 5, $\mathcal{L}_{SUP}$ is maximum likelihood loss, $D_{tr}$ is a labelled training dataset of $(x, y)$ pairs, and $\lambda$ is a hyperparameter.

---

[1] IUNET prunes an ineffective supernetwork into an efficient effective subnetwork. OMP prunes an inefficient effective supernetwork into an efficient but slightly less effective subnetwork.

[2] Proof of Theorem 1 provided in Appendix.

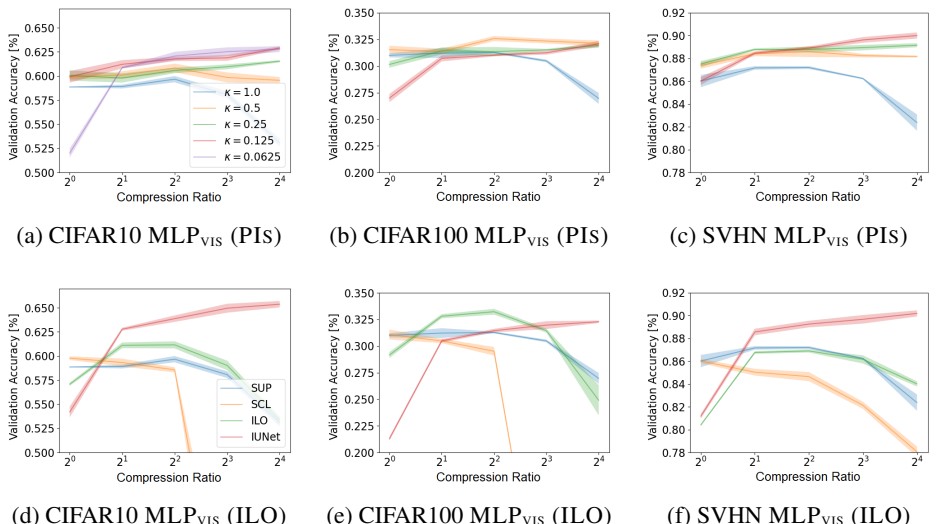

Figure 2: Effect of PIs and ILO on pruned models. The y-axis is the validation accuracy (%) and x-axis is the compression ratio. PIs experiments only alter the supernetwork's initialization. $\kappa = 1.0$ means normal initialization. ILO experiments only alter the training objective during supernetwork training. After supernetwork training, subnetworks are pruned under different compression ratios, then finetuned. Validation accuracy of trained pruned models are reported.

$$\mathcal{L}(\theta_M; \mathcal{S}) = \mathbb{E}_{x,y \sim D_{tr}} \left[ \mathcal{L}_{SUP}(x, y, \theta_M) + \lambda \mathcal{L}_{NCE}(x, y, \theta_M; \mathcal{S}) \right] \tag{6}$$

### 3.2.2 PROACTIVE INITIALIZATION SCHEME: PIS

Deep neural networks often enter the lazy training regime (Chizat et al., 2019; Liu et al., 2023), where the loss steadily decreases while weights barely change. This is particularly harmful to neural networks pruning (Liu et al., 2023), especially when low-magnitude weights contribute to decreasing the loss and hence should not be pruned.

We propose a simple solution by scaling the weight initialization by a small multiplier, $\kappa$. We find this alleviates the aforementioned issue by forcing the model to assign large values only to important weights prior to lazy training. Because lazy training is only an issue for pruning, we only apply $\kappa$-scaling to the pre-pruning training stage, not the fine-tuning stage. This is done by scaling the initial weights $\theta_M^{(0)} = \kappa \theta_{M\dagger}^{(0)}$, where $\theta_{M\dagger}^{(0)}$ follows the Kaiming (He et al., 2015) or Glorot (Glorot & Bengio, 2010) initialization.

## 4 EXPERIMENT SETUP

### 4.1 DATASETS

IUNET is evaluated on *image* and *tabular* classification [3]:

- **Vision**: Experiments are run on CIFAR10, CIFAR100, and SVHN (Krizhevsky et al., 2009; Netzer et al., 2011), following baseline work (Neyshabur, 2020)[4].

- **Tabular**: Experiments are run on 40 tabular datasets from a benchmark paper (Kadra et al., 2021), covering a diverse range of problems. The datasets were collected from OpenML (Gijsbers et al., 2019), UCI (Asuncion & Newman, 2007), and Kaggle.

---

[3]More details are provided in the Supplementary.

[4]While SMC benchmark (Liu et al., 2023) is open-sourced, the code is being cleaned-up at submission time.

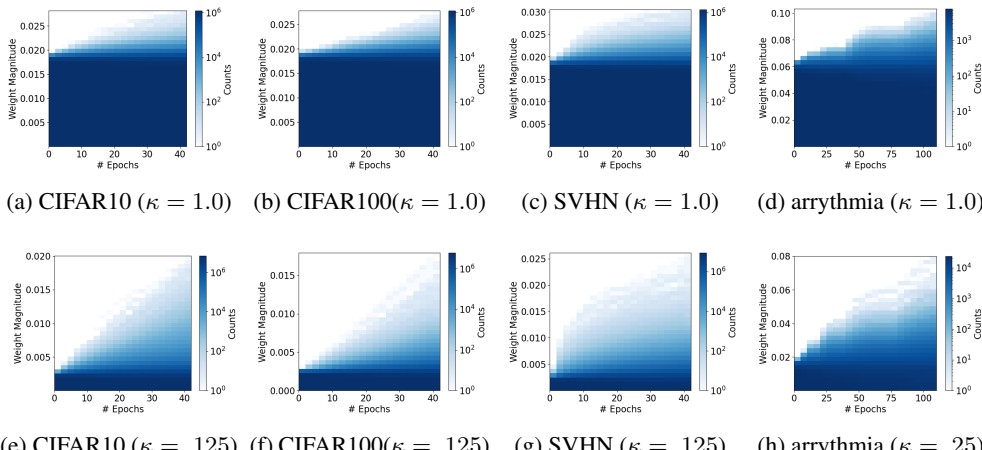

(a) CIFAR10 ($\kappa = 1.0$)  (b) CIFAR100($\kappa = 1.0$)  (c) SVHN ($\kappa = 1.0$)  (d) arrythmia ($\kappa = 1.0$)

(e) CIFAR10 ($\kappa = .125$)  (f) CIFAR100($\kappa = .125$)  (g) SVHN ($\kappa = .125$)  (h) arrythmia ($\kappa = .25$)

Figure 3: Histogram of weight magnitudes, $|\theta_M^{(t)}|$, plotted over each epoch under different $\kappa$ initializations settings. $\kappa = 1.0$ means normal initialization. Results shown for MLP$_{\text{VIS}}$ on the CIFAR10, CIFAR100, and SVHN datasets.

## 4.2 MODEL SETUP

IUNET is compared against One-shot Magnitude Pruning (OMP) (Blalock et al., 2020), and $\beta$-LASSO pruning (Neyshabur, 2020) on all datasets. We denote the supernetwork used by each pruning method with a superscript. Unless otherwise specified, models are trained via maximum likelihood. In addition, we compare against the following dataset-specific supernetworks (MLP$_{\text{VIS}}$, MLP$_{\text{TAB}}$, RESNET) and models:

- **Vision**: We consider RESNET (He et al., 2016), MLP$_{\text{VIS}}$, a MLP that contains a CNN subnetwork (Neyshabur, 2020), and the aforementioned CNN subnetwork.

- **Tabular**: We consider MLP$_{\text{TAB}}$, a 9-layer MLP with hidden dimension 512 (Kadra et al., 2021), XGB (Chen & Guestrin, 2016), TABN (Arik & Pfister, 2021), a handcrafted tabular deep learning architecture, and MLP$_{\text{TAB+C}}$ (Kadra et al., 2021), the state-of-the-art MLP, which was heavily tuned from a cocktail of regularization techniques.

## 4.3 CONSIDERED INVARIANCES

The success of contrastive learning on both vision and tabular datasets indicates their corresponding invariant transformations, $\mathcal{S}$, are desirable for each task. For computer vision, SimCLR (Chen et al., 2020) transformations are used: (1) resize crops, (2) horizontal flips, (3) color jitter, and (4) random grayscale. For tabular learning, SCARF (Bahri et al., 2021) transformations are used: (5) randomly corrupting features by drawing the corrupted versions from its empirical marginal distribution.

## 5 RESULTS

### 5.1 ON INDUCTIVE BIAS

In this section, we compare the effectiveness of the trained subnetwork discovered by IUNET, $f^P(\cdot; \theta_P^{(T)})$, against the trained supernetwork, $f^M(\cdot; \theta_M^{(T)})$. As seen in Tables 1 and 7, the pruned subnetwork outperforms the original supernetwork, even though the supernetwork has more representational capacity. This supports our claim that IUNET prunes subnetwork architectures with better inductive biases than the supernetwork. Importantly, IUNET substantially improves upon existing pruning baselines by explicitly including invariances via ILO and alleviating the lazy learning issue (Liu et al., 2023) via PIS.

On *vision* datasets: As seen in Table 1, IUNET is a general and flexible framework that improves the inductive bias of not only models like MLP$_{\text{VIS}}$ but also specialized architectures like RESNET. Specifcally, IUNET $^{(\text{MLP}_{\text{VIS}})}$ bridges the gap between MLPs and CNNs. Unlike previous work (Tolstikhin et al., 2021), IUNET $^{(\text{MLP}_{\text{VIS}})}$ does this in an entirely automated procedure. IUNET $^{(\text{RESNET})}$ achieves the best performance, indicating IUNET can be applied across various models.

On *tabular* datasets: As seen in Table 2, the subnetworks derived from MLPs outperform both the original MLP$_{\text{TAB}}$ and hand-crafted architectures: TABN and XGB. Unlike vision, how to encode invariances for tabular data is highly nontrivial, making IUNET particularly effective. The gains made by MLP$_{\text{TAB}}$ is similar to those from MLP$_{\text{TAB+C}}$ (Kadra et al., 2021), which ran extensive hyperparameter tuning on top of MLP$_{\text{TAB}}$. Unlike MLP$_{\text{TAB+C}}$, IUNET requires substantially less time tuning hyperparameters. Note, IUNET $^{(\text{MLP}_{\text{TAB}})}$ did not use the optimal hyperparameters found by MLP$_{\text{TAB+C}}$. Furthermore, because IUNET is a flexible framework, it can be combined with new models/ trainig techniques on tabular data as they are discovered.

## 5.2 ABLATION STUDY

To study the effectiveness of (1) pruning, (2) PIs, and (3) ILO, each one is removed from the optimal model. As seen in Table 4, each is crucial to IUNET. Pruning is necessary to encode the inductive bias into the subnetwork's neural architecture. PIs and ILO improves the pruning policy by ensuring weights crucial to finetuning and capturing invariance are not pruned. Notice, without pruning, IUNET $_{\text{NO-PRUNE}}$ performs worse than the original supernetwork. This highlights an important notion that PIs aims to improve the pruning policy, not the unpruned performance. By sacrificing unpruned performance, PIs ensures important weights are not falsely pruned. PIs is less effective on tabular datasets where the false pruning issue seems less severe. Combining pruning, ILO, and PIs, IUNET most consistently achieves the best performance.

## 5.3 EFFECTS OF PRUNING

To further study the effects of pruning, we plot how performance changes over different compression ratios. Figure 2 clearly identifies how PIs and ILO substantially improves upon existing pruning policies. First, our results support existing findings that (1) OMP does not produce subnetworks that substantially outperform the supernetwork (Blalock et al., 2020) and (2) while unpruned models trained with SCL can outperform supervised ones, pruned models trained with SCL perform substantially worse (Corti et al., 2022). PIs flips the trend from (1) by slightly sacrificing unpruned performance, due to poorer initialization, IUNET discovers pruned models with better inductive biases, which improves downstream performance. ILO fixes the poor performance of SCL in (2) by preserving information pathways for both invariance and max likelihood over training. We highlight both these findings are significant among the network pruning community. Finally, Figure 2 confirms IUNET achieves the best performance by combining both PIs and ILO.

In addition to being more effective that the supernetwork, $f^M(\cdot; \theta_M^{(T)})$, the pruned network, $f^P(\cdot; \theta_P^{(T)})$, is also more efficient. Figure 2 shows IUNET can reach 8-16$\times$ compression while still keeping superior performance.

## 5.4 EFFECT OF PROACTIVE INITIALIZATION

To further study the role of PIs, the histogram of weight magnitudes is monitored over the course of training. As shown in Figure 3, under the standard OMP pruning setup, the histogram changes little over the course of training, which supports the lazy training hypothesis (Liu et al., 2023) where performance rapidly improves, while weight magnitudes change very little, decoupling each weight's importance from its magnitude.

With PIs, only important weights grow over the course of training, while most weights remain near zero, barely affecting the output activations of each layer. This phenomenon alleviates the lazy training problem by ensuring (1) pruning safety, as pruned weights are near zero prior which have minimal affect on layer activations, and (2) importance-magnitude coupling, as structurally important connections must grow to affect the output of the layer.

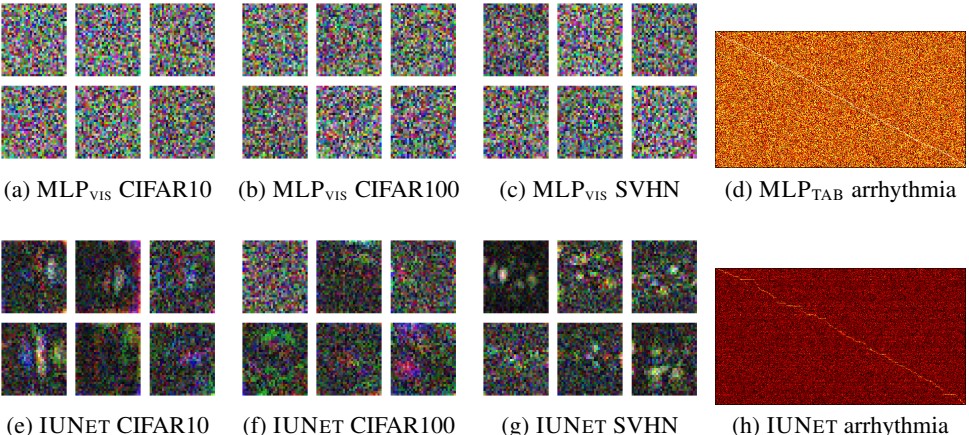

(a) MLP$_{\text{VIS}}$ CIFAR10     (b) MLP$_{\text{VIS}}$ CIFAR100     (c) MLP$_{\text{VIS}}$ SVHN     (d) MLP$_{\text{TAB}}$ arrhythmia

(e) IU$_{\text{NET}}$ CIFAR10     (f) IU$_{\text{NET}}$ CIFAR100     (g) IU$_{\text{NET}}$ SVHN     (h) IU$_{\text{NET}}$ arrhythmia

Figure 4: Visualization of weight magnitudes, $|\theta_M^{(T)}|$, trained with different policies. The top row was trained on CIFAR10 and shows the magnitude of each RGB pixel for 6 output logits. The bottom row was trained on arrhythmia and shows the weight matrix of the 1st layer with 280 input and 512 output dimensions. Lighter color means larger magnitude.

## 5.5 ON INVARIANCE CONSISTENCY

To further study whether particular invariances are learned, we compute the consistency metric (Singla et al., 2021), which measure the percentage of samples whose predicted label would flip when an invariant transformation is applied to the input. As seen in Table 3, the subnetwork found by IUNET, $f^P(\cdot; \theta_P^{(0)})$, is able to preserve invariances specified in ILO much better than the supernetwork, $f^M(\cdot; \theta_M^{(0)})$. This shows IUNET indeed captures desirable invariances.

## 5.6 ON WEIGHT VISUALIZATION

We visualize the supernetwork weights, $\theta_M^{(T)}$, when trained with IUNET compared to standard maximum likelihood (MLP) to determine what structures preserve invariance.

On *vision* datasets: As seen in Figure 4, IUNET learns more locally connected structures, which improves translation invariance. Prior work (Neyshabur, 2020) found network structure (as opposed to inductive bias) to be the limiting factor for encoding CNN inductive biases into MLPs, which IUNET successfully replicates.

On *tabular* datasets: As seen in Figure 4, IUNET weights focus more on singular features. This preserves invariance over random feature corruption, as the absence of some tabular features does not greatly alter output activations of most neurons. This structure can also be likened to tree ensembles (Grinsztajn et al., 2022), whose leaves split individual features rather than all features.

## 6 CONCLUSION

In this work, we study the viability of network pruning for discovering invariant-preserving architectures. Under the computer vision setting, IUNET bridges the gap between deep MLPs and deep CNNs, and reliably boosts RESNET performance. Under the tabular setting, IUNET reliably boosts performance of existing MLPs, comparable to applying the state-of-the-art regularization cocktails. Our proposed novelties, ILO and PIs, flexibly improves existing OMP pruning policies by both successfully integrating contrastive learning and alleviating lazy training. Thus, IUNET effectively uses pruning to tackle invariance learning.

## 7 REPRODUCIBILITY STATEMENT

We provide a complete description of the data processing steps in Section 4.1 and Appendix E.1. We cover hyperparameters used in Section 4.2, Section 4.3, Appendix E.2, and Appendix E.3. We cover pruning implementation details in Appendix E.4. We cover hardware and approximate runtime in Appendix E.5. The proof for Theorem 1 can be found in Appendix B.1.

## 8 ETHICS STATEMENT

There are no new datasets released by this work, hence it did not involve human subjects. Datasets used in this work were adopted from existing benchmarks (Neyshabur, 2020; Blalock et al., 2020; Kadra et al., 2021), as described in Section 4.1 and Appendix E.1. There are no harms introduced by this work. This work aims to improve both effectiveness and efficiency of representation learning.

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

# A  ADDITIONAL RELATED WORK

## A.1  TABULAR MACHINE LEARNING

Tabular data is a difficult regime for deep learning, where deep learning models struggle against decision tree approaches. Early methods use forests, ensembling, and boosting (Shwartz-Ziv & Armon, 2022; Borisov et al., 2022; Chen & Guestrin, 2016). Later, researchers handcrafted new deep architectures that mimic trees (Popov et al., 2019; Katzir et al., 2020; Huang et al., 2020; Hazimeh et al., 2020; Somepalli et al., 2021; Arik & Pfister, 2021). Yet, when evaluated on large datasets, these approaches are still beaten by XGB (Chen & Guestrin, 2016; Grinsztajn et al., 2022). Recent work found MLPs with heavy regularization tuning (Kadra et al., 2021) can outperform decision tree approaches, though this conclusion does not hold on small tabular datasets (Joseph & Raj, 2022). To specially tackle the small data regime, Bayesian learning and Hopfield networks are combined with MLPs (Hollmann et al., 2022; Schäfl et al., 2022). There are also work on tabular transformers (Huang et al., 2020; Gorishniy et al., 2021), though said approaches require much more training data. Without regularization, tree based models still outperform MLPs due to a better inductive bias and resilience to noise (Grinsztajn et al., 2022). Different data preprocessing level encodings are being proposed to boost MLP performance (Gorishniy et al., 2022). To the best of our knowledge, the state-of-the-art on general tabular datasets remain heavily regularized MLPs ($MLP_{TAB+C}$) (Kadra et al., 2021; Gorishniy et al., 2022). We aim to further boost regularized MLP performance by discovering model architectures that capture good invariances from tabular data.

## A.2  CONTRASTIVE LEARNING

Contrastive learning, initially proposed for metric learning (Chopra et al., 2005; Schroff et al., 2015; Oh Song et al., 2016), trains a model to learn shared features among images of the same type (Jaiswal et al., 2020). It has been widely used in self-supervised pretraining (Chen et al., 2020; 2021), where dataset augmentation is crucial. Although contrastive learning was originally proposed for images, it has also shown promising results in graph data (Zhu et al., 2021; You et al., 2020b), speech data (Baevski et al., 2020), and tabular data (Bahri et al., 2021). Previous study has showed that speech transformers tend to overfit the contrastive loss in deeper layers, suggesting that removing later layers can be beneficial during finetuning (Pasad et al., 2021). While contrastive learning performs well pretraining unpruned models, its vanilla formulation performs poorly after network pruning (Corti et al., 2022). In this work, we establish a connection between contrastive learning and invariance learning and observe that pruned contrastive models fail because of overfitting.

## A.3  NEURAL ARCHITECTURE SEARCH

Neural Architecture Search (NAS) explores large superarchitectures by leveraging smaller block architectures (Wan et al., 2020; Pham et al., 2018; Zoph et al., 2018; Luo et al., 2018; Liu et al., 2018a). These block architectures are typically small convolutional neural networks (CNNs) or MLPs. The key idea behind NAS is to utilize these blocks (Pham et al., 2018; Zoph et al., 2018) to

capture desired invariance properties for downstream tasks. Prior works (You et al., 2020a; Xie et al., 2019) have analyzed randomly selected intra- and inter-block structures and observed performance differences between said structures. However, these work did not propose a method for discovering block architectures directly from data. Our work aims to address this gap by focusing on discovering the architecture within NAS blocks. This approach has the potential to enable NAS in diverse domains, expanding its applicability beyond the current scope.

## B  LOSS FUNCTION DETAILS

We provide a more detailed description of our loss function in this section. Following notation from the main paper, we repeat the ILO loss function in Equation 7 below:

$$\mathcal{L}(\theta; \mathcal{S}) = \mathbb{E}_{x,y \sim D_{train}} \left[ \mathcal{L}_{SUP}(x, y, \theta) + \lambda \mathcal{L}_{NCE}(x, y, \theta; \mathcal{S}) \right] \tag{7}$$

To better explain our loss functions, we introduce some new notations. First, we denote the decoder output probability function over classes, $\mathcal{Y}$, as $\tilde{p}_{\theta_{\mathcal{D}}} : \mathcal{H} \to [0, 1]^{|\mathcal{Y}|}$, where $f_{\mathcal{D}} = \text{argmax} \circ \tilde{p}_{\theta_{\mathcal{D}}}$. We denote the model output probability function by combining $\tilde{p}_{\theta_{\mathcal{D}}}$ with the encoder as follows: $p_{\theta} = \tilde{p}_{\theta_{\mathcal{D}}} \circ f_{\mathcal{E}}$. We introduce an integer mapping from classes $\mathcal{Y}$ as $\mathcal{I} : \mathcal{Y} \to \{0, 1, 2, ..., |\mathcal{Y}| - 1\}$.

We show the maximum likelihood loss, $\mathcal{L}_{SUP}$, in Equation 8 below.

$$\mathcal{L}_{SUP}(x, y, \theta) = -log(p_{\theta}(x, \theta)_{\mathcal{I}(y)}) \tag{8}$$

We show the supervised contrastive loss, $\mathcal{L}_{NCE}$, in Equation 9 below. Following SimCLR (Chen et al., 2020), we assume that the intermediary representations are $d$-dimensional embeddings, $\mathcal{H} = \mathbb{R}^d$, and use the cosine similarity as our similarity function, $\psi^{(cos)} : \mathbb{R}^d \times \mathbb{R}^d \to \mathbb{R}$.

$$\mathcal{L}_{NCE}(x, y, \theta; \mathcal{S}) = \mathbb{E}_{g \sim \mathcal{S}} \left[ -log \left( \frac{exp\left(\psi^{(cos)}\left(f_{\mathcal{E}}(x; \theta), f_{\mathcal{E}}(g(x); \theta)\right)\right)}{\sum_{\substack{x', y' \sim D_{tr} \\ y' \neq y}} exp\left(\psi^{(cos)}\left(f_{\mathcal{E}}(x; \theta), f_{\mathcal{E}}(g(x'); \theta)\right)\right)} \right) \right] \tag{9}$$

### B.1  SURROGATE OBJECTIVE

We aim to learn invariance-preserving network architectures from the data. In our framework, this involves optimizing our invariance objective, which we repeat in Equation 10. We prove Theorem 1, that minimizing the supervised contrastive loss is equivalent to maximizing the invariance objective, outlined below.

$$\theta^* = \underset{\theta}{argmax} \underset{\substack{x_i, x_j \sim \mathcal{X} \\ g \sim \mathcal{S}}}{\mathbb{E}} \left[ \frac{\phi(f_{\mathcal{E}}^M(x_i; \theta), f_{\mathcal{E}}^M(x_j; \theta))}{\phi(f_{\mathcal{E}}^M(x_i; \theta), f_{\mathcal{E}}^M(g(x_i); \theta))} \right] \tag{10}$$

We convert the distance metric $\phi$ into similarity metric $\psi$.

| Dataset | $g(\cdot)$ | $\text{MLP}_{\text{VIS}}$ | $\text{IUNET}^{(\text{MLP}_{\text{VIS}})}$ |
|---|---|---|---|
| CIFAR10 | resize. | $44.096 \pm 0.434$ | $\mathbf{97.349 \pm 4.590}$ |
| | horiz. | $80.485 \pm 0.504$ | $\mathbf{99.413 \pm 1.016}$ |
| | color. | $56.075 \pm 0.433$ | $\mathbf{98.233 \pm 3.060}$ |
| | graysc. | $81.932 \pm 0.233$ | $\mathbf{99.077 \pm 1.598}$ |
| CIFAR100 | resize. | $32.990 \pm 1.065$ | $\mathbf{39.936 \pm 2.786}$ |
| | horiz. | $70.793 \pm 0.677$ | $\mathbf{77.935 \pm 1.464}$ |
| | color. | $31.704 \pm 0.560$ | $\mathbf{51.397 \pm 2.709}$ |
| | graysc. | $71.245 \pm 0.467$ | $\mathbf{76.476 \pm 1.245}$ |
| SVHN | resize. | $36.708 \pm 2.033$ | $\mathbf{77.440 \pm 0.627}$ |
| | horiz. | $71.400 \pm 1.651$ | $\mathbf{95.082 \pm 0.166}$ |
| | color. | $61.341 \pm 0.946$ | $\mathbf{91.097 \pm 0.395}$ |
| | graysc. | $90.344 \pm 0.233$ | $\mathbf{99.259 \pm 0.073}$ |

Table 5: Comparing the consistency metric (%) of the untrained supernetwork, $\text{MLP}_{\text{VIS}}$ and $\text{MLP}_{\text{TAB}}$, against IUNET's pruned subnetwork under different invariant transforms, $g(\cdot)$. IUNET preserves invariances better.

$$
\begin{aligned}
\theta^* &= \underset{\theta}{argmax} \underset{\substack{x_i,x_j \sim \mathcal{X} \\ g \sim \mathcal{S}}}{\mathbb{E}} \left[ \frac{\psi(f_{\mathcal{E}}^M(x_i;\theta), f_{\mathcal{E}}^M(g(x_i);\theta))}{\psi(f_{\mathcal{E}}^M(x_i;\theta), f_{\mathcal{E}}^M(x_j;\theta))} \right] \\
&= \underset{\theta}{argmin} \underset{\substack{x_i,x_j \sim \mathcal{X} \\ g \sim \mathcal{S}}}{\mathbb{E}} \left[ \frac{-\psi(f_{\mathcal{E}}^M(x_i;\theta), f_{\mathcal{E}}^M(g(x_i);\theta))}{\psi(f_{\mathcal{E}}^M(x_i;\theta), f_{\mathcal{E}}^M(x_j;\theta))} \right] \\
&= \underset{\theta}{argmin} \underset{\substack{x \sim \mathcal{X} \\ g \sim \mathcal{S}}}{\mathbb{E}} \left[ \frac{-\psi(f_{\mathcal{E}}^M(x;\theta), f_{\mathcal{E}}^M(g(x);\theta))}{\sum\limits_{\substack{x' \sim \mathcal{X} \\ x' \neq x}} \psi(f_{\mathcal{E}}^M(x;\theta), f_{\mathcal{E}}^M(g(x');\theta))} \right] \\
&= \underset{\theta}{argmin} \underset{\substack{x,y \sim D_{tr} \\ g \sim \mathcal{S}}}{\mathbb{E}} \left[ \frac{-\psi\left(f_{\mathcal{E}}^M(x;\theta), f_{\mathcal{E}}^M(g(x);\theta)\right)}{\sum\limits_{\substack{x',y' \sim D_{tr} \\ y' \neq y}} \psi\left(f_{\mathcal{E}}^M(x;\theta), f_{\mathcal{E}}^M(g(x');\theta)\right)} \right] \\
&= \underset{\theta}{argmin} \underset{\substack{x,y \sim D_{tr} \\ g \sim \mathcal{S}}}{\mathbb{E}} \left[ -log \left( \frac{\psi\left(f_{\mathcal{E}}^M(x;\theta), f_{\mathcal{E}}^M(g(x);\theta)\right)}{\sum\limits_{\substack{x',y' \sim D_{tr} \\ y' \neq y}} \psi\left(f_{\mathcal{E}}^M(x;\theta), f_{\mathcal{E}}^M(g(x');\theta)\right)} \right) \right]
\end{aligned}
\tag{11}
$$

We set the similarity metric, $\psi$, to be the same as our contrastive loss: $\psi(\cdot) = exp(\psi^{(cos)}(\cdot))$.

$$
\theta^* = \underset{\theta}{argmin} \underset{x,y \sim D_{train}}{\mathbb{E}} \left[ \mathcal{L}_{NCE}(x,y,\theta;\mathcal{S}) \right]
\tag{12}
$$

Here, we showed that the vanilla contrastive loss function, Equation 9, serves as a surrogate objective for optimizing our desired invariance objective, Equation 10. By incorporating contrastive learning alongside the maximum likelihood objective in Equation 7, ILO effectively reveals the underlying invariances in the pruned model.

## C  ADDITIONAL DISCUSSION ON LAZY TRAINING

The lazy training regime (Chizat et al., 2019; Tzen & Raginsky, 2020) is a phenomenon when loss rapidly decreases, while weight values stay relatively constant. This phenomenon occurs on large

| Dataset | $\text{MLP}_{\text{VIS}}$ | $\text{IUNET}^{(\text{MLP}_{\text{VIS}})}$ | CNN | $\text{IUNET}^{(\text{RESNET})}$ |
|---|---|---|---|---|
| CIFAR10 | $59.266 \pm 0.050$ | $64.847 \pm 0.121$ | $75.850 \pm 0.788$ | $\mathbf{83.729 \pm 0.153}$ |
| CIFAR100 | $31.052 \pm 0.371$ | $32.760 \pm 0.288$ | $41.634 \pm 0.402$ | $\mathbf{53.099 \pm 0.243}$ |
| SVHN | $84.463 \pm 0.393$ | $89.357 \pm 0.156$ | $91.892 \pm 0.411$ | $\mathbf{94.020 \pm 0.291}$ |

Table 6: Comparing the pruned $\text{IUNET}^{(\text{MLP}_{\text{VIS}})}$ model to a CNN which is architecturally equivalent to $\text{IUNET}^{(\text{MLP}_{\text{VIS}})}$ in terms of layer count and hidden dimensions with the only differences being network structure and weight sharing. Although $\text{IUNET}^{(\text{MLP}_{\text{VIS}})}$ cannot outperform CNN, it bridges the gap between MLP and CNN architectures without any human design intervention.

over-parameterized neural networks (Chizat et al., 2019). Because the weight values stay relatively constant, the magnitude ordering between weights also changes very little. Therefore, network pruning struggles to preserve such loss decreases in the lazy training regime (Liu et al., 2023).

Because weights with very small magnitude have minimal effect on the output logits, pruning said weights will not drastically hurt performance. Thus, if the pruning framework can separate very small magnitude weights from normal weights prior to the lazy training regime, we can preserve loss decreases in the lazy training regime. The PIS setting accomplishes this by initializing all weights to be very small so that only important weights will learn large magnitudes. This guarantees that a large percentage of weights will have small magnitudes throughout training, while important larger magnitude weights will emerge over the course of training.

# D   ADDITIONAL EXPERIMENTS

## D.1   ON TABULAR DATASETS: FULL RESULTS

We provide the full tabular dataset results in Table 7. As shown, the trends reported in the main text holds on the whole dataset.

## D.2   ON CONSISTENCY: FULL RESULTS

We provide consistency experiments on CIFAR100 and SVHN in Table 5. As shown, the trends reported in the main text holds on other datasets.

## D.3   COMPARING IUNET WITH CNN

We compare $\text{IUNET}^{(\text{MLP}_{\text{VIS}})}$ with a CNN in Table 6. Unlike $\text{IUNET}^{(\text{MLP}_{\text{VIS}})}$, CNNs also employ weight sharing. While IUNET consistently improves performance of both $\text{MLP}_{\text{VIS}}$ and RESNET via pruning the model architecture, exploration of weight sharing is an orthogonal direction that could also reduce the gap between MLPs and CNNs. Note, $\text{IUNET}^{(\text{RESNET})}$ still performs the best out of all models.

# E   IMPLEMENTATION DETAILS

## E.1   DATASET DETAILS

We considered the following *computer vision* datasets: CIFAR10, CIFAR100 (Krizhevsky et al., 2009), and SVHN (Netzer et al., 2011). CIFAR10 and CIFAR100 are multi-domain image classification datasets. SVHN is a street sign digit classification dataset. Input images are $32 \times 32$ color images. We split the train set by 80/20 for training and validation. We test on the test set provided separately. We reported dataset statistics in Table 8.

We considered 40 tabular datasets from OpenML (Gijsbers et al., 2019), UCI (Asuncion & Newman, 2007), and Kaggle, following the $\text{MLP}_{\text{TAB+C}}$ benchmark (Kadra et al., 2021). These tabular datasets cover a variety of domains, data types, and class imbalances. We used a 60/20/20 train validation test split, and reported dataset statistics in Table 9. We use a random seed of 11 for the data split, following prior work (Kadra et al., 2021).

| Dataset | MLP$_{TAB}$ | OMP | $\beta$-Lasso | IUNet | XGB | TabN | MLP$_{TAB+C}$ |
|---|---|---|---|---|---|---|---|
| credit-g | 70.000 | 70.000 | 67.205 | 63.166 ± 0.000 | 68.929 | 61.190 | **74.643** |
| anneal | 99.490 | 99.691 | 99.634 | **99.712 ± 0.101** | 85.416 | 84.248 | 89.270 |
| kr-vs-kp | 99.158 | 99.062 | 99.049 | 99.151 ± 0.064 | **99.850** | 93.250 | **99.850** |
| arrhythmia | 67.086 | 55.483 | 67.719 | **74.138 ± 2.769** | 48.779 | 43.562 | 61.461 |
| mfeat. | 98.169 | 97.959 | 97.204 | **98.176 ± 0.121** | 98.000 | 97.250 | 98.000 |
| vehicle | 80.427 | 81.115 | 80.611 | 81.805 ± 2.065 | 74.973 | 79.654 | **82.576** |
| kc1 | 80.762 | **84.597** | 83.587 | **84.597 ± 0.000** | 66.846 | 52.517 | 74.381 |
| adult | 81.968 | 82.212 | 82.323 | 78.249 ± 3.085 | 79.824 | 77.155 | **82.443** |
| walking. | 58.466 | 60.033 | 58.049 | 59.789 ± 0.456 | 61.616 | 56.801 | **63.923** |
| phoneme | 84.213 | 86.733 | 84.850 | 87.284 ± 0.436 | **87.972** | 86.824 | 86.619 |
| skin-seg. | 99.869 | 99.866 | 99.851 | 99.876 ± 0.006 | **99.968** | 99.961 | 99.953 |
| ldpa | 66.590 | 68.458 | 62.362 | 64.816 ± 4.535 | **99.008** | 54.815 | 68.107 |
| nomao | 95.776 | 95.682 | 95.756 | 95.703 ± 0.110 | **96.872** | 95.425 | 96.826 |
| cnae | 94.080 | 92.742 | 94.808 | **96.075 ± 0.242** | 94.907 | 89.352 | 95.833 |
| blood. | 68.965 | 61.841 | 65.126 | **70.375 ± 5.255** | 62.281 | 64.327 | 67.617 |
| bank. | **88.300** | **88.300** | 86.923 | **88.300 ± 0.000** | 72.658 | 70.639 | 85.993 |
| connect. | 72.111 | 72.016 | 72.400 | 74.475 ± 0.445 | 72.374 | 72.045 | **80.073** |
| shuttle | 99.709 | 93.791 | 99.687 | 93.735 ± 2.303 | 98.563 | 88.017 | **99.948** |
| higgs | 72.192 | 72.668 | 72.263 | 73.215 ± 0.384 | 72.944 | 72.036 | **73.546** |
| australian | 82.153 | 83.942 | 81.667 | 82.562 ± 1.927 | **89.717** | 85.278 | 87.088 |
| car | 99.966 | **100.000** | **100.000** | 99.859 ± 0.200 | 92.376 | 98.701 | 99.587 |
| segment | 91.504 | 91.603 | 91.317 | 91.563 ± 0.000 | **93.723** | 91.775 | **93.723** |
| fashion. | 91.139 | 90.784 | 90.864 | 90.817 ± 0.040 | 91.243 | 89.793 | **91.950** |
| jungle. | 86.998 | 92.071 | 87.400 | 95.130 ± 0.807 | 87.325 | 73.425 | **97.471** |
| numerai | 51.621 | 51.443 | 51.905 | 51.839 ± 0.067 | 52.363 | 51.599 | **52.668** |
| devnagari | 97.550 | 97.573 | 97.549 | 97.517 ± 0.014 | 93.310 | 94.179 | **98.370** |
| helena | 29.342 | 28.459 | 29.834 | **29.884 ± 0.991** | 21.994 | 19.032 | 27.701 |
| jannis | 68.647 | 66.302 | 69.302 | **69.998 ± 1.232** | 55.225 | 56.214 | 65.287 |
| volkert | 70.066 | 68.781 | 69.655 | 70.104 ± 0.215 | 64.170 | 59.409 | **71.667** |
| miniboone | 86.539 | 87.575 | 87.751 | 81.226 ± 6.569 | **94.024** | 62.173 | 94.015 |
| apsfailure | 97.041 | **98.191** | 98.048 | **98.191 ± 0.000** | 88.825 | 51.444 | 92.535 |
| christine | 70.295 | 69.819 | 70.275 | 69.065 ± 1.225 | **74.815** | 69.649 | 74.262 |
| dilbert | 98.494 | 98.738 | 98.522 | 98.540 ± 0.023 | **99.106** | 97.608 | 99.049 |
| fabert | 65.540 | 64.709 | 66.681 | 65.695 ± 0.065 | **70.098** | 62.277 | 69.183 |
| jasmine | 78.691 | 80.139 | 78.415 | **80.864 ± 0.374** | 80.546 | 76.690 | 79.217 |
| sylvine | 92.660 | 92.650 | 92.593 | 93.369 ± 0.833 | **95.509** | 83.595 | 94.045 |
| dionis | 93.920 | 93.687 | 93.943 | 93.586 ± 0.021 | 91.222 | 83.960 | **94.010** |
| aloi | 96.546 | 96.376 | 96.562 | 95.341 ± 0.194 | 95.338 | 93.589 | **97.175** |
| ccfraud | 97.554 | 97.748 | 96.626 | **98.797 ± 1.031** | 90.303 | 85.705 | 92.531 |
| clickpred. | 82.175 | 83.206 | 82.307 | **85.270 ± 1.275** | 58.361 | 50.163 | 64.280 |

Table 7: Comparing IUNet against trees (XGB), handcrafted models (TabN), and state-of-the-art regularized MLPs (MLP$_{TAB+C}$). OMP, $\beta$-Lasso, and IUNet all modify MLP$_{TAB}$. Note, our method does not tune the optimal regularization settings for each dataset making it more efficient. Our pruned model is also more compressed than the original network. Note, we outperform both MLP$_{TAB}$ and TabN on most datasets. While IUNet performs similarly to MLP$_{TAB+C}$, it does not require costly hyperparameter tuning, and can be applied on top of the optimal settings found by MLP$_{TAB+C}$.

### E.2 Hyperparameter Settings

All experiments were run 3 times from scratch starting with different random seeds. We report both the mean and standard deviation of all runs. All hyperparameters were chosen based on validation set results.

For all experiments, we used $\lambda = 1$, which was chosen through a grid search over $\lambda \in \{0.25, 0.5, 1.0\}$. For all experiments, we used a batch size of 128. For pre-pruning training, we used SGD with Nesterov momentum and a learning rate of 0.001, following past works (Blalock et al., 2020). For finetuning vision datasets, we used the same optimizer setup except with 16-bit operations except for batch normalization, following $\beta$-Lasso (Neyshabur, 2020). For finetuning tabular datasets, we used AdamW (Loshchilov & Hutter, 2017), a learning rate of 0.001s, decoupled weight decay, cosine annealing with restart, initial restart budget of 15 epochs, budget multiplier of 2, and snapshot ensembling (Huang et al., 2017), following prior works (Kadra et al., 2021; Zimmer et al., 2021). It is important to note we did not tune the dataset and training hyperparameters for each tabular dataset individually like $\text{MLP}_{\text{TAB+C}}$ (Kadra et al., 2021), rather taking the most effective setting on average.

For tabular datasets, we tuned the compression ratio over the following range of values: $r \in \{2, 4, 8\}$ and the PIs multiplier over the following range of values: $\kappa \in \{0.25, 0.125, 0.0625\}$. on a subset of 4 tabular datasets. We found that $r = 8$ and $\kappa = 0.25$ performs the most consistently and used this setting for all runs of IUNET in the main paper. It is important to note we did not tune hyperparameters for IUNET on each individual tabular dataset like $\text{MLP}_{\text{TAB+C}}$ (Kadra et al., 2021), making IUNET a much more efficient model than $\text{MLP}_{\text{TAB+C}}$. For the tabular baselines (Chen & Guestrin, 2016; Arik & Pfister, 2021; Kadra et al., 2021), we used the same hyperparameter tuning setup as the MLP+C benchmark (Kadra et al., 2021).

For vision datasets, we tuned the compression ratio over the following range of values: $r \in \{2, 4, 8, 16\}$ on each individual dataset for all network pruning models except $\beta$-Lasso[5]. For $\beta$-lasso (Neyshabur, 2020), we tuned the hyperparameters over the range $\beta = \{50\}$ and L1 regularization in $l1 \in \{10^{-6}, 2 \times 10^{-6}, 5 \times 10^{-6}, 10^{-5}, 2 \times 10^{-5}\}$ on each individual dataset as done in the original paper. It is important to note that although we tuned both hyperparameters for both IUNET and baselines on each individual datasets, our main and ablation table rankings stay consistent had we chosen a single setting for all datasets, as shown in the detailed pruning experiments in the main paper.

### E.3 Supernetwork Architecture

$\text{MLP}_{\text{VIS}}$ is a deep MLP that contains a CNN subnetwork. Given a scaling factor, $\alpha$, the CNN architecture consists of 3x3 convolutional layers with the following (out channels, stride) settings: $[(\alpha, 1), (2\alpha, 2), (2\alpha, 1), (4\alpha, 2), (4\alpha, 1), (8\alpha, 2), (8\alpha, 1), (16\alpha, 2)]$ followed by a hidden layer of dimension $64\alpha$. It is worth noting that our CNN does not include maxpooling layers for fair comparison with the learned architectures, following the same setup as $\beta$-Lasso (Neyshabur, 2020). To form the MLP Network, we ensured the CNN network structure exists as a subnetwork within the MLP supernetwork by setting the hidden layer sizes to: $[\alpha s^2, \frac{\alpha s^2}{2}, \frac{\alpha s^2}{2}, \frac{\alpha s^2}{4}, \frac{\alpha s^2}{4}, \frac{\alpha s^2}{8}, \frac{\alpha s^2}{8}, \frac{\alpha s^2}{16}, 64\alpha]$. This architecture was also introduced in $\beta$-Lasso (Neyshabur, 2020). All layers are preceded by batch normalization and ReLU activation. We chose $\alpha = 8$ such that our supernetwork can fit onto an Nvidia RTX 3070 GPU.

CNN is the corresponding CNN subnetwork with (out channels, stride) settings: $[(\alpha, 1), (2\alpha, 2), (2\alpha, 1), (4\alpha, 2), (4\alpha, 1), (8\alpha, 2), (8\alpha, 1), (16\alpha, 2)]$, derived from prior works (Neyshabur, 2020). Again, we chose $\alpha = 8$ to be consistent with $\text{MLP}_{\text{VIS}}$.

RESNET (He et al., 2016) is the standard RESNET-18 model used in past benchmarks (Blalock et al., 2020). Resnet differs from CNN in its inclusion of max-pooling layers and residual connections.

$\text{MLP}_{\text{TAB}}$ is a 9-layer MLP with hidden dimension 512, batch normalization, and ReLU activation. We did not use dropout or skip connections as it was found to be ineffective on most tabular datasets in MLP+C (Kadra et al., 2021).

---

[5]This is because $\beta$-Lasso does not accept a chosen compression ratio as a hyperparameter.

| Dataset | # Train Instances | # Valid Instances | # Test Instances | Number of Classes |
|---------|-------------------|-------------------|------------------|-------------------|
| CIFAR10 | 40000 | 10000 | 10000 | 10 |
| CIFAR100 | 40000 | 10000 | 10000 | 100 |
| SVHN | 58606 | 14651 | 26032 | 10 |

Table 8: Statistics on computer vision datasets.

### E.4 PRUNING IMPLEMENTATION DETAILS

Following Shrinkbench (Blalock et al., 2020), we use magnitude-based pruning only on the encoder, $f_{\mathcal{E}}$, keeping all weights in the decoder, $f_{\mathcal{D}}$. This is done to prevent pruning a cutset in the decoder architecture, so that all class logits receive input signal. To optimize the performance, we apply magnitude-based pruning globally, instead of layer-wise.

### E.5 HARDWARE

All experiments were conducted on an Nvidia V100 GPU and an AMD EPYC 7402 CPU. The duration of the tabular experiments varied, ranging from a few minutes up to half a day, depending on the specific dataset-model pair and the training phase (pre-pruning training or finetuning). For the vision experiments, a single setting on a single dataset-model pair required a few hours for both pre-pruning training and finetuning.

| Dataset | # Train Inst. | # Valid Inst. | # Test Inst. | # Feats. | Majority Class % | Minority Class % | OpenML ID |
|---|---|---|---|---|---|---|---|
| Anneal | 538 | 179 | 179 | 39 | 76.17 | 0.89 | 233090 |
| Kr-vs-kp | 1917 | 639 | 639 | 37 | 52.22 | 47.78 | 233091 |
| Arrhythmia | 271 | 90 | 90 | 280 | 54.20 | 0.44 | 233092 |
| Mfeat-factors | 1200 | 400 | 400 | 217 | 10.00 | 10.00 | 233093 |
| Credit-g | 600 | 200 | 200 | 21 | 70.00 | 30.00 | 233088 |
| Vehicle | 507 | 169 | 169 | 19 | 25.77 | 23.52 | 233094 |
| Kc1 | 1265 | 421 | 421 | 22 | 84.54 | 15.46 | 233096 |
| Adult | 29305 | 9768 | 9768 | 15 | 76.07 | 23.93 | 233099 |
| Walking-activity | 89599 | 29866 | 29866 | 5 | 14.73 | 0.61 | 233102 |
| Phoneme | 3242 | 1080 | 1080 | 6 | 70.65 | 29.35 | 233103 |
| Skin-segmentation | 147034 | 49011 | 49011 | 4 | 79.25 | 20.75 | 233104 |
| Ldpa | 98916 | 32972 | 32972 | 8 | 33.05 | 0.84 | 233106 |
| Nomao | 20679 | 6893 | 6893 | 119 | 71.44 | 28.56 | 233107 |
| Cnae-9 | 648 | 216 | 216 | 857 | 11.11 | 11.11 | 233108 |
| Blood-transfusion | 448 | 149 | 149 | 5 | 76.20 | 23.80 | 233109 |
| Bank-marketing | 27126 | 9042 | 9042 | 17 | 88.30 | 11.70 | 233110 |
| Connect-4 | 40534 | 13511 | 13511 | 43 | 65.83 | 9.55 | 233112 |
| Shuttle | 34800 | 11600 | 11600 | 10 | 78.60 | 0.02 | 233113 |
| Higgs | 58830 | 19610 | 19610 | 29 | 52.86 | 47.14 | 233114 |
| Australian | 414 | 138 | 138 | 15 | 55.51 | 44.49 | 233115 |
| Car | 1036 | 345 | 345 | 7 | 70.02 | 3.76 | 233116 |
| Segment | 1386 | 462 | 462 | 20 | 14.29 | 14.29 | 233117 |
| Fashion-MNIST | 42000 | 14000 | 14000 | 785 | 10.00 | 10.00 | 233118 |
| Jungle-Chess-2pcs | 26891 | 8963 | 8963 | 7 | 51.46 | 9.67 | 233119 |
| Numerai28.6 | 57792 | 19264 | 19264 | 22 | 50.52 | 49.48 | 233120 |
| Devnagari-Script | 55200 | 18400 | 18400 | 1025 | 2.17 | 2.17 | 233121 |
| Helena | 39117 | 13039 | 13039 | 28 | 6.14 | 0.17 | 233122 |
| Jannis | 50239 | 16746 | 16746 | 55 | 46.01 | 2.01 | 233123 |
| Volkert | 34986 | 11662 | 11662 | 181 | 21.96 | 2.33 | 233124 |
| MiniBooNE | 78038 | 26012 | 26012 | 51 | 71.94 | 28.06 | 233126 |
| APSFailure | 45600 | 15200 | 15200 | 171 | 98.19 | 1.81 | 233130 |
| Christine | 3250 | 1083 | 1083 | 1637 | 50.00 | 50.00 | 233131 |
| Dilbert | 6000 | 2000 | 2000 | 2001 | 20.49 | 19.13 | 233132 |
| Fabert | 4942 | 1647 | 1647 | 801 | 23.39 | 6.09 | 233133 |
| Jasmine | 1790 | 596 | 596 | 145 | 50.00 | 50.00 | 233134 |
| Sylvine | 3074 | 1024 | 1024 | 21 | 50.00 | 50.00 | 233135 |
| Dionis | 249712 | 83237 | 83237 | 61 | 0.59 | 0.21 | 233137 |
| Aloi | 64800 | 21600 | 21600 | 129 | 0.10 | 0.10 | 233142 |
| C.C.FraudD | 170884 | 56961 | 56961 | 31 | 99.83 | 0.17 | 233143 |
| Click Prediction | 239689 | 79896 | 79896 | 12 | 83.21 | 16.79 | 233146 |

Table 9: Statistics on tabular datasets. Note that the OpenML ID denotes the ID used to retrieve the dataset (Gijsbers et al., 2019). Majority and Minority Class % shows the class imbalance within each dataset. For fair evaluation, we report balanced accuracy in all tabular experiments. # Feats. denotes the number of features in each dataset.

