# OpenReview forum: "Learning Invariances via Neural Network Pruning"
_ICLR.cc/2024/Conference — Submitted to ICLR 2024_

### Official Review · Reviewer_iziS · 2023-11-01

**Soundness:** 2 fair
**Presentation:** 2 fair
**Contribution:** 3 good
**Rating:** 5
**Confidence:** 3

**Summary:**

The paper proposes a pruning framework called Invariance Unveiling Neural Networks (IUNET) that automatically discovers invariance-preserving subnetworks from deep and dense supernetworks. The framework combines pruning with a novel invariance learning objective to improve the network's inductive bias. In particular, the training objective minimizes the distance between representations of inputs under invariant perturbations for the supernetwork such that the pruned subnetwork architecture captures the invariance. The authors demonstrate that their approach outperforms existing methods in terms of efficiency and effectiveness on both vision and tabular datasets.

**Strengths:**

The strengths of the paper lie in its comprehensive exploration of invariance in neural network architectures, its demonstration of improved performance on various datasets, and its proposal of a training objective that considers both maximum likelihood and desired invariance properties. The paper also addresses the challenges in designing invariant neural networks for tabular data, which is a valuable contribution to the field.

**Weaknesses:**

- As far as I understand, how connectivity models achieve invariance / equivariance does not only rely on the architecture but also on certain constraints on weights (especially weight sharing), for example, the translation invariance / equivariance with convolution. This work only emphasizes on the architecture, but there are no discussions on weights. I am doubtful of the claim (hypothesis) that the subnetwork after retraining "preserves desired invariances and hence improves the inductive bias" and I wonder if this is empirically the case.

- There lacks a justification or analysis on how pruning of architecture is able to achieve for certain transformation groups. The mechanisms of transformations on the values of input tensors (e.g. colour) should apparently be different from the transformations on the domain (e.g. translation and rotation).

- Rotation, as one of the most explored transformation in the literature of invariant/equivariant deep learning, is however not included in the study of this work.

- A couple of details in the experimental setup are not clear enough, and those will highly affect understanding the effectiveness of the proposed method. See questions.

**Questions:**

-  After pruning, is ILO or data augmentation used in training the subnetwork?

- In table 3, are the weights of pruned IUNets also re-initialised or re-trained before evaluating the consistency? And if re-trained is ILO or augmentation used in training? As I mentioned, the inductive bias of invariance / equivariance does not only exist in the architecture but also in the shared weights. So if the IUNet weights are re-initialised then the results are too good to be true, otherwise an unfair comparison.

- Also in table 3 and 4, what transformations are used?

- In table 4, what if data augmentation is applied to all models, no matter with and without ILO? I wonder how ILO can improve over simple augmentation (especially with a pruned model trained without ILO)

- Why in fig2(e) ILO is better than IUNet?

---

> ### Author Response · Authors · 2023-11-21
> **Rebuttal Response Part 1**
>
> Thank you for your insightful comments! We provided additional experiments and clarifications to address your concerns. Please let us know if you have any remaining doubts!
>
> **[On Architectural Emphasis]**
>
> Our goal is to improve the **network architecture’s inductive bias**.
>
> We agree weights play an important role in preserving invariance; however, the **network architecture’s inductive bias** is **independent of its weights** and refers to the inductive bias captured by **the structure of the network**. Improving the network architecture’s structure will improve its prior and optimization trajectory[1] **leading to better trained models**. In fact convolutional nets can be used as feature extractors **even under random initialization**[1,2,3]. Learning invariance by solely updating the weights will not achieve a better architectural inductive bias.
>
> We empirically verified IUNet’s pruned network has **a better structure and inductive bias in our ablation studies, Table 4.** Our results show the pruned subnetwork achieves **substantially better** downstream accuracy than both the supernetwork trained normally and the supernetwork trained with our novelties, PIs and ILO.
>
> It is *“empirically the case” IUNet’s captures better such invariances. As detailed in the [On Consistency Experiment] rebuttal section, network pruning is *“necessary for obtaining an invariance-preserving network”* under maximum likelihood training.
>
> We will update our main text to emphasize our motivation for improving the **network architecture’s inductive bias** rather than just the weights.
>
> **[On Consistency Experiment 1/2]**
>
> The weights are re-initialized to right after the first epoch fairly for all models in Table 3.
>
> We greatly appreciate your suggestion to deeply re-analyze these results. We found an edge case that the consistency metric did not properly capture. Although IUNet’s consistency metric is >98%, **the model predicts the same class for nearly all inputs**! This is because IUNet has majority small weights and not been fully trained, so the encoder consistently predicts a near-zero vector representation, regardless of input image.
>
> **To correct for this edge case, we rerun Table 3 and additional consistency experiments on CIFAR10, CIFAR100, and SVHN.** Because IUNet lacks weight sharing, we expect randomly-initialized CNN to outperform both it and MLPs. With maximum likelihood training, we expect IUNet to effectively bridge the invariance consistency gap between MLPs and CNNs, by learning CNN-like weights from its improved inductive bias. Overall, we expect IUNet to outperform MLPs in accuracy by learning a network structure with better inductive bias.

---

> > ### Author Response · Authors · 2023-11-21
> > **Rebuttal Response Part 2**
> >
> > **[On Consistency Experiment 2/2]**
> >
> > As per request, we tested randomly **Kaiming-initialized** models, MLP, IUNet, and CNN, on the **consistency metric**, but re-emphasize both the weights and architecture needed to fully quantify the inductive bias. Because weights are re-initialized to the same range for all models, the “same class” issue no longer occurs.
> >
> > ```
> > CIFAR10 (randomly initialized)
> > Method          | resize.  | horiz.   | color.   | graysc.
> > —---------------+----------+----------+----------+----------
> > MLP             | 23.2±1.2 | 38.8±0.7 | 42.4±1.9 | 78.2±0.5
> > IUNet           | 23.0±1.6 | 39.2±1.9 | 42.8±1.7 | 77.7±0.7
> > CNN             | 30.1±1.3 | 57.4±1.5 | 52.3±2.7 | 85.0±1.2
> >
> > CIFAR100 (randomly initialized)
> > Method          | resize.  | horiz.   | color.   | graysc.
> > —---------------+----------+----------+----------+----------
> > MLP             | 12.1±1.0 | 28.1±1.6 | 29.3±1.3 | 68.5±0.3
> > IUNet           | 12.0±1.6 | 28.6±1.3 | 29.9±1.4 | 68.6±0.7
> > CNN             | 12.7±2.1 | 45.3±1.7 | 33.8±0.8 | 77.2±2.3
> >
> > SVHN (randomly initialized)
> > Method          | resize.  | horiz.   | color.   | graysc.
> > —---------------+----------+----------+----------+----------
> > MLP             | 26.2±1.4 | 38.5±2.2 | 42.8±1.7 | 79.3±0.7
> > IUNet           | 30.2±4.7 | 42.7±4.3 | 45.8±3.3 | 80.9±1.7
> > CNN             | 40.2±5.6 | 60.4±3.1 | 53.8±3.0 | 86.5±2.0
> > ```
> >
> > In the second experiment, we tested the **fully trained** models, MLP, IUNet (before pruning), IUNet, and CNN, on the **consistency metric**. Note, IUNet (before pruning) is equivalent to an MLP trained with the proposed PIs initialization and the ILO learning objective. This is unlike MLP, IUNet and CNN, which were all trained with maximum likelihood. Because models are fully trained, the “same class” issue no longer occurs.
> >
> > ```
> > CIFAR10 (fully-trained)
> > Method          | resize.  | horiz.   | color.   | graysc.
> > —---------------+----------+----------+----------+----------
> > MLP             | 71.8±0.5 | 86.1±0.6 | 77.0±0.2 | 87.8±0.3
> > IUNet(no-prune) | 77.0±0.3 | 88.9±0.1 | 88.7±0.4 | 90.4±0.2
> > IUNet           | 75.3±0.3 | 88.7±0.1 | 81.9±0.2 | 89.9±0.2
> > CNN             | 77.0±0.3 | 88.1±0.4 | 87.3±0.1 | 94.4±0.6
> >
> > CIFAR100 (fully-trained)
> > Method          | resize.  | horiz.   | color.   | graysc.
> > —---------------+----------+----------+----------+----------
> > MLP             | 55.5±0.4 | 80.4±0.2 | 54.4±0.3 | 78.1±0.2
> > IUNet(no-prune) | 63.7±0.4 | 82.9±0.2 | 73.3±0.8 | 81.9±0.2
> > IUNet           | 58.9±0.5 | 79.8±0.1 | 60.6±0.5 | 78.8±0.4
> > CNN             | 60.8±1.4 | 78.4±0.2 | 66.1±1.3 | 82.6±0.8
> >
> > SVHN (fully-trained)
> > Method          | resize.  | horiz.   | color.   | graysc.
> > —---------------+----------+----------+----------+----------
> > MLP             | 78.3±0.6 | 89.5±0.2 | 93.5±0.4 | 98.2±0.1
> > IUNet(no-prune) | 80.4±0.2 | 88.2±0.0 | 93.7±0.2 | 98.6±0.0
> > IUNet           | 83.8±0.6 | 92.8±0.2 | 95.1±0.1 | 99.0±0.0
> > CNN             | 87.2±1.3 | 93.5±1.3 | 95.4±1.4 | 99.1±0.4
> > ```
> >
> > In the third experiment, we tested the **fully trained** models, MLP, IUNet (before pruning), IUNet, and CNN, on **accuracy** (Table 4, ablation studies).
> >
> >  ```
> > Method          | CIFAR10  | CIFAR100 | SVHN
> > —---------------+----------+----------+----------
> > MLP             | 59.266   | 31.052   | 84.463
> > IUNet(no-prune) | 54.622   | 20.332   | 78.427
> > IUNet           | 64.847   | 32.760   | 89.357
> > CNN             | 75.850   | 41.634   | 91.892
> > ```
> >
> > **We make the following conclusions from our updates results:**
> >
> > 1) **Under random initialization:** IUNet preserves invariance at least as well as MLPs. CNNs preserve more invariance than both IUNet and MLP due to weight sharing.
> >
> > 2) **Role of invariance:** Invariance of the untrained model does not reliably predict downstream performance, because weights play an important role in the trained model. Under different training setups (maximum likelihood v.s. PIs + ILO), invariance of the trained model does not predict downstream accuracy. **Under the same training setup (maximum likelihood), the invariance of the trained model predicts downstream accuracy.**
> >
> > 3) **Under maximum likelihood training:** IUNet preserves more invariance than MLPs but less than CNNs, bridging the gap between MLPs and CNNs.
> >
> > 4) **Role of Pruning:** It is unclear whether IUNet trained with maximum likelihood preserves more invariance than MLPs trained with PIs+ILO (IUNet (no-prune)), but IUNet trained with maximum likelihood consistently outperforms IUNet (no-prune) in downstream accuracy. Hence, **pruning retains invariance preserving structures, such that a pruned model trained solely on maximum likelihood will preserve invariance learned in ILO-optimized weights.**
> >
> > 5) **Role of PIs and ILO:** PIs+ILO effectively learn weights that preserve invariance empirically.
> >
> > Again, we thank you for exposing this important detail. We will replace Table 3’s results with these updated results in our main paper.

---

> > > ### Author Response · Authors · 2023-11-21
> > > **Rebuttal Response Part 3**
> > >
> > > **[On Justification for How Pruning Achieves Group Transformations]**
> > >
> > > How pruning achieves group transformations can be best seen in Figure 4. In Figure 4, we visualize randomly-chosen filters from the first layer of each network. We randomly chose the filters for fairness, and picked the first layer because it is the easiest to interpret.
> > >
> > > The main conclusion from Figure 4 is: **IUNet learns architectures similar to CNNs and decision trees from dense MLPs**. We make the following specific observations:
> > >
> > > **1) Locally Connected Regions:** For Figures 4efg, IUNet finds locally connected regions with high RGB values. Local connections help preserve translation and scale invariance as shown in Table 3 and verified by previous works[1,2,3,4]. The presence of locally-connected regions visually confirms that IUNet can rediscover CNN architectures from MLPs.
> > >
> > > **2) Color Invariance:** In Figures 4eg (CIFAR10/SVHN), IUNet assigns high values to locally connected regions uniformly across all RGB channels, which improves color invariance. In Figure 4f (CIFAR100), this is not as consistent due to a more challenging learning objective, though we find it still improves color invariance over the MLP supernetwork.
> > >
> > > **3) Discovering Substructures:** Nearly all IUNet output neurons favor a clear pattern of input neurons (the exception being the top left filter of Figure 4f), whereas nearly no OMP output neurons exhibit any pattern. This suggests IUNet does a good job recovering invariance preserving substructures across the supernetwork’s first layer.
> > >
> > > **4) Axis-Aligned Tree Structures:** For Figure 4h, IUNet discovers subnetworks that focus on primarily 1 input feature for each output neuron. This behavior mirrors tree-based models where neurons are activated where single features reach some threshold [5,6,7]. Such network architecture follows existing work which found the axis-aligned inductive bias in tree-based models is critical to XGBoost’s performance over standard MLP setups [7].
> > >
> > > These visualization show IUNet automatically prunes MLPs to *“preserve desired invariances”* (such as ones found in CNNs) *“and hence improves the inductive bias”*.
> > >
> > > **[On Rotation Invariance]**
> > >
> > > We did not introduce rotations into our pipeline since the **SVHN dataset is a digit classification** task. Specifically, digits are not rotationally invariant since 6 and 9 fall under the same space group yet are distinct digits.
> > >
> > > **[On Transformations and Data Augmentations]**
> > >
> > > In Tables 3 and 4, the set of transformations used are mentioned in Section 4.3. For computer vision, these are (1) resize crops, (2) horizontal flips, (3) color jitter, and (4) random grayscale. For tabular data, this is (5) SCARF[8], randomly corrupting features by drawing the corrupted versions from its empirical marginal distribution.
> > >
> > > In Tables 3 and 4, all methods adopt some **standard dataset augmentation**. On vision datasets, we follow ShrinkBench[9], which adopts random horizontal flips and crops. On tabular datasets, we follow MLP+C[4], which adopts no augmentation by default.
> > >
> > > **For fairness, the maximum likelihood branch is trained on the same dataset, with or without ILO.** The contrastive learning branch applies transformations (1-5) on training samples to create positive and negative pairs, but the underlying data used to train all models are the same. We will add these details to Section E.
> > >
> > > **[On Figure 2]**
> > >
> > > As seen in Figure 2abc, **PIs** will tradeoff poorer performance in the unpruned network for better performance in the pruned network, **thus becoming more effective as the sparsity increases**. In Figure 2def, IUNet combines ILO and PIs, hence at lower sparsities, ILO can outperform IUNet, but as sparsity increases, IUNet begins outperforming ILO.

---

> > > > ### Author Response · Authors · 2023-11-21
> > > > **Rebuttal Response Part 4**
> > > >
> > > > **[References]**
> > > >
> > > > [1] Ulyanov, Dmitry, Andrea Vedaldi, and Victor Lempitsky. "Deep image prior." Proceedings of the IEEE conference on computer vision and pattern recognition. 2018.
> > > >
> > > > [2] Cao, Yun-Hao, and Jianxin Wu. "A random cnn sees objects: One inductive bias of cnn and its applications." Proceedings Of The AAAI Conference On Artificial Intelligence. Vol. 36. No. 1. 2022.
> > > >
> > > > [3] Rosenfeld, Amir, and John K. Tsotsos. "Intriguing properties of randomly weighted networks: Generalizing while learning next to nothing." 2019 16th conference on computer and robot vision (CRV). IEEE, 2019.
> > > >
> > > > [4] Neyshabur, Behnam. "Towards learning convolutions from scratch." Advances in Neural Information Processing Systems 33 (2020): 8078-8088.
> > > >
> > > > [5] Marton, Sascha, et al. "GRANDE: Gradient-Based Decision Tree Ensembles." arXiv preprint arXiv:2309.17130 (2023).
> > > >
> > > > [6] Marton, Sascha, et al. "GradTree: Learning Axis-Aligned Decision Trees with Gradient Descent." NeurIPS 2023 Second Table Representation Learning Workshop. 2023.
> > > >
> > > > [7] Grinsztajn, Léo, Edouard Oyallon, and Gaël Varoquaux. "Why do tree-based models still outperform deep learning on typical tabular data?." Advances in Neural Information Processing Systems 35 (2022): 507-520.
> > > >
> > > > [8] Bahri, Dara, et al. "Scarf: Self-supervised contrastive learning using random feature corruption." arXiv preprint arXiv:2106.15147 (2021).
> > > >
> > > > [9] Blalock, Davis, et al. "What is the state of neural network pruning?." Proceedings of machine learning and systems 2 (2020): 129-146.

---

> > > > > ### Author Response · Authors · 2023-11-22
> > > > > **Gentle Reminder on Discussion Deadline**
> > > > >
> > > > > Dear Reviewer iziS,
> > > > >
> > > > > Thank you again for your insightful comments. Based on your suggestions, we spent much effort in experiments and writing to address your concerns. As the deadline for discussion nears, are there any additional questions or concerns you still have regarding our work? We are eager to help with questions or concerns.

---

> > > > > > ### Comment · Reviewer_iziS · 2023-11-22
> > > > > >
> > > > > > Thank the reviewers for very detailed response. A number of my concerns are well addressed but not all of them.
> > > > > >
> > > > > > Specifically, I hold the opinion that for a variety of equivariance, inductive priors can be built on the architecture for two aspects: connectivity and weight sharing (e.g. Cohen and Welling. Group Equivariant Convolutional Networks), while pruning can only address the former, but not the latter. As an example, it is not enough to achieve ConvNet structure from MLPs merely from pruning the connections to be local, because the weight sharing patterns of the sliding window filter also plays an important role. These two aspects together make convnets a good inductive prior for images.
> > > > > >
> > > > > > Another related concern is that the addressed types of transforms g() are too simple to stress test the ability and limitation of applying pruning methods to learning invariance. More challenging transforms should be involved, e.g. rotation, which is the most explored one in the invariant/equivariant learning literature. The authors did not include such an experiment because in the SVHN dataset there will be ambiguities with "6" and "9", but this is very easy to overcome by just removing one class from the dataset.

---

> > > > > > > ### Author Response · Authors · 2023-11-23
> > > > > > > **On the Two Aspects of Inductive Priors**
> > > > > > >
> > > > > > > Thank you for your timely reply! We would like to clarify why our work focuses on connectivity.
> > > > > > >
> > > > > > > Our focus on connectivity is motivated by existing work[1] which found **connectivity is more important than weight sharing w.r.t. CNN's inductive prior**: *"the gap between a locally connected network and its corresponding fully-connected version is much higher than that between convolutional [(locally-connected + weight sharing)] and locally-connected networks"*. Hence, while the ConvNet structure includes both connectivity and weight sharing, **the bottleneck in ConvNet's inductive prior is its connectivity.**
> > > > > > >
> > > > > > > IUNet automatically discovers aforementioned connectivity from data, boosting multiple supernetworks' performances across multiple domains via pruning. We highlight IUNet is **the first to systematically improve connectivity across multiple deep supernetworks/domains via pruning.** Thus, while we leave weight sharing and rotation invariance for future work, we believe IUNet is an important addition to the existing invariance and equivariance learning literature.
> > > > > > >
> > > > > > > We hope this context is helpful and will update our paper to make it more apparent.
> > > > > > >
> > > > > > > Please let us know any other questions you may have! Thanks!
> > > > > > >
> > > > > > > [1] Neyshabur, Behnam. "Towards learning convolutions from scratch." Advances in Neural Information Processing Systems 33 (2020): 8078-8088.

---

### Official Review · Reviewer_LdpC · 2023-11-02

**Soundness:** 3 good
**Presentation:** 2 fair
**Contribution:** 3 good
**Rating:** 5
**Confidence:** 5

**Summary:**

This paper aims to address the issue of pruning neural networks (MLP) which affects the accuracy of the new model, it mainly addresses the issue of invariance during pruning (meaning the weights which might have learned the invariance during the training are dropped) which affects the model accuracy after typical pruning on transformed or original data. It also uses techniques of Initialization of weights and carefully dropping the weights during pruning. They combine Contrastive loss and Maximum likelihood loss to finetune the subnetworks which eventually outperform the original model.

**Strengths:**

- The paper has a really good notion of using existing work to initialize the weights, dropping the weights and even reinitializing the weights after pruning.

- An automated procedure the construct a completely new sub-network.

- They discover invariance-preserving sub-networks that outperform the invariance-agnostic supernetworks, removing inductive bias in Super net is addressed.

- Paper claims really good improvement on MLP after Pruning when the transformation is performed which fills the gap between CNN and MLP performance.

**Weaknesses:**

- Most of the ideas are derived from different papers and combined together.

- Some errors in annotations, missing graph line for different values of K, repetition of the words and techniques throughout the paper.

- The annexure table 7 for tabular data is a bit unclear as it highlights only a few good instances, while the other pruning methods still work better than the proposed method. The reason is not mentioned. While the idea is to capture and preserve the invariance during the pruning, this means somewhere something is still missing in the proposed method, it solves the problem in some of the datasets but not all.

- The paper mentions the encoder-decoder architecture of the proposed method, but throughout the paper, the convention is not followed properly.

- The paper aims to learn an architecture which creates an encoder which represents the same encoding values for all types of transformation.

**Questions:**

- The authors mention that their method improves compression performance on existing models, how about a completely new model? Does it still work the same? Or do the pre-trained models need to be trained extensively?

---

> ### Author Response · Authors · 2023-11-21
> **Rebuttal Response Part 1**
>
> Thank you for your response! We provided additional experiments and clarifications to address your concerns. Please let us know if you have any remaining doubts!
>
> **[On Compressing Pretrained vs Randomly Initialized Networks]**
>
> Our experiments are run on **randomly initialized networks** (i.e. *“completely new models”*) using **“existing” architectures** (CNNs and MLPs). IUNet (1) **randomly initializes** the supernetwork with weights $\theta_{M^\dagger}^{(0)}$, (2) scales the random weights with PIs, $\theta_M^{(0)} = \kappa \theta_{M^\dagger}^{(0)}$, (3) trains the supernetwork **from scratch** using ILO to get $\theta_M^{(T)}$, (4) prunes the trained supernetwork into a subnetwork, (5) **randomly re-initializes** the pruned subnetwork as $\theta_{P}^{(0)}$ with Lottery Ticket Reinitialization (i.e. resetting weights to that of $\theta_{M^\dagger}^{(0)}$), and (6) finetunes the **randomly re-initialized** subnetwork **from scratch** using maximum likelihood to get $\theta_P^{(T)}$.
>
> Yes, **IUNet works on pretrained networks with minimal adaptation!** We provide **new experiments** on **pretrained transformers**, following the SMC-Benchmark[1], which prunes the pretrained RoBERTa[2] on the Common Sense QA dataset[3]. Using the official Github repo, *https://github.com/VITA-Group/SMC-Bench/*, we reran SMC’s provided pruning algorithms OMP (after), OMP (before), and SNIP[4]. Additionally, we implemented and ran IUNet and GRaSP[5] on this benchmark. We report the accuracy under different sparsities.
>
> For a fair comparison, we leave out ILO from IUNet and adopt the same finetuning protocol and hyperparameters as SMC. We tried 2 different settings for kappa: [0.125, 0.0625]. Because SMC uses pretrained models: (1) IUNet scales the pretrained weights rather than random initialization by kappa, (2) IUNet reinitializes the pruned models weights to the fine-tuned model initialized from pretrained weights, not random initializations.
>
> Note, IUNet is a **post-tuning pruning algorithm** (similar to OMP (after) in the benchmark paper), where the model is first pretrained, then fine-tuned (either with maximum likelihood or ILO), then pruned, then sparsely fine-tuned to convergence. OMP(before), SNIP, and GRaSP are **pruning at initialization (PaI)**[1, 6] algorithms, where the model first pretrained, then pruned (iterating through the train set once) then sparsely fine-tuned to convergence.
>
> ```
> Method\Sparsity   | 0.2   | 0.36  | 0.488 | 0.590 | 0.672
> —-----------------+-------+-------+-------+-------+-------
> OMP (Before)      | 26.54 | 58.89 | 25.88 | 19.57 | 19.57
> SNIP              | 18.67 | 20.48 | 18.84 | 19.00 | 19.16
> GRaSP             | 21.79 | 19.25 | 18.43 | 17.77 | 20.63
> —-----------------+-------+-------+-------+-------+-------
> OMP (After)       | 76.09 | 73.87 | 30.14 | 19.57 | 19.57
> —-----------------+-------+-------+-------+-------+-------
> IUNet* (k=0.125)  | 75.10 | 74.53 | 34.89 | 19.57 | 19.57
> IUNet* (k=0.0625) | 75.02 | 74.37 | 53.80 | 19.57 | 19.57
> *no-ILO
> ```
>
> The densely fine-tuned model reaches an accuracy of 75.76%.
>
> At low sparsities (sparsity= [0.2, 0.36]), **all post-tuning pruning algorithms (OMP(After) and IUNet) outperform PaI algorithms (OMP(Before), SNIP, GRaSP)**. At these sparsities, *OMP(After) and IUNet are within experimental uncertainty of each other*. This agrees with prior work[1, 6] which finds PaI is less effective than post-tuning algorithms.
>
> At middle sparsities (sparsity=0.488), **IUNet outperforms all baselines (OMP (after), OMP (before), SNIP, GRaSP),** because PIs alleviates the lazy learning phenomenon which is a key bottleneck in existing pruning algorithms[4]. This is a notable finding, showing **IUNet scales to larger sparsities than baselines.**
>
> At high sparsities (sparsity= [0.5, 0.672]), **all methods fail**, which agrees with SMC[4], suggesting at least 50% of weights are crucial to the transformer models on these tasks.
>
> **Hence, as shown in results above, IUNet generalizes to pre-trained models and outperforms strong baselines.**

---

> ### Author Response · Authors · 2023-11-21
> **Rebuttal Response Part 2**
>
> **[On Novelty and Existing Work]**
>
> We highlight our contributions over the existing works and will update our writing to make this more clear.
>
> On task novelty:
>
> **IUNet is the first general-purpose invariance learning via pruning approach**. Our approach tackles vision, text, and tabular data across MLPs, CNNs, and transformers. Previous work[8] only explored computer vision where the authors state they hope to *“see if [network pruning for invariance learning] can be used in other contexts”*.
>
>
> On method novelty:
>
> 1) ILO: **We are the first to combine maximum likelihood (cross entropy) with contrastive learning (NCE) for network pruning. This combination improves network pruning, which contradicts previous work[7] that found contrastive learning hurts network pruning.** The authors of [7] claim they hope their negative results *“spark interest in developing more representation-friendly pruning methods for supervised contrastive learning”.*  We successfully demonstrate previous efforts failed because they overfit the contrastive objective[14]. Combining both maximum likelihood and contrastive learning is necessary to boost pruning performance.
>
> 2) PIs: **We are the first to propose small weight initialization for network pruning.** Small initial weights force the neural network to only grow weights that are truly important. Otherwise, unimportant weights will stay large, causing important weights to be pruned (i.e. lazy training phenomenon[1, 15]). This issue occurs more on deeper networks[15]. Hence, unlike earlier invariance learning for pruning work[8], **IUNet improves the scalability of said approach to deep networks**.
>
> **[On Tabular Dataset Results]**
> The competing baseline “MLP+C” performs time-consuming (few days for all 40 datasets) hyperparameter optimization over regularization techniques, whereas our approach does not (few hours for all 40 datasets). Thus, there is a trade-off between MLP+C's more time-consuming good results compared to IUNet’s more economical, yet still quite good results.
>
> It is well known that tabular data has tricky results as the dataset quality can drastically vary: from sparse to dense inputs and from few to many samples[9, 10, 11, 12]. Some datasets are naturally biased towards more sample efficient algorithms like XGBoost[10, 13]. Other datasets may be bottlenecked by regularization rather than architectural design[9, 11, 12]. Despite this, **IUNet boosts MLP performance on tabular data in aggregate and achieves the best performance without expensive hyperparameter tuning.**
>
> **[Clarifying Figure 2]**
>
> Thank you for pointing out this errata! We will updated the paper with the correct x- and y-labels. Results with kappa=0.0625 are too poor (below the visible y-axis). We will update our figures to clarify this as well.
>
> **[Clarifying Encoder and Decoder Blocks]**
>
> Thank you for pointing out this detail! We will update Section 3.2 to be consistent. In terms of empirical results, as mentioned in Appendix E.4, we follow the encoder-decoder architecture, where only the encoder is pruned and the decoder (final linear layer following conventions) remains dense.
>
> **[Learning different Pruning Masks for different transformation types]**
>
> We thank the reviewer for this excellent suggestion, but leave this as future work.

---

> > ### Author Response · Authors · 2023-11-21
> > **Rebuttal Response Part 3**
> >
> > **[References]**
> >
> > [1] Liu, Shiwei, et al. "Sparsity May Cry: Let Us Fail (Current) Sparse Neural Networks Together!." arXiv preprint arXiv:2303.02141 (2023).
> >
> > [2] Liu, Yinhan, et al. "Roberta: A robustly optimized bert pretraining approach." arXiv preprint arXiv:1907.11692 (2019).
> >
> > [3] Talmor, Alon, et al. "Commonsenseqa: A question answering challenge targeting commonsense knowledge." arXiv preprint arXiv:1811.00937 (2018).
> >
> > [4] Lee, Namhoon, Thalaiyasingam Ajanthan, and Philip HS Torr. "Snip: Single-shot network pruning based on connection sensitivity." arXiv preprint arXiv:1810.02340 (2018).
> >
> > [5] Wang, Chaoqi, Guodong Zhang, and Roger Grosse. "Picking winning tickets before training by preserving gradient flow." arXiv preprint arXiv:2002.07376 (2020).
> >
> > [6] Frankle, Jonathan, et al. "Pruning neural networks at initialization: Why are we missing the mark?." arXiv preprint arXiv:2009.08576 (2020).
> >
> > [7] Corti, Francesco, et al. "Studying the impact of magnitude pruning on contrastive learning methods." arXiv preprint arXiv:2207.00200 (2022).
> >
> > [8] Neyshabur, Behnam. "Towards learning convolutions from scratch." Advances in Neural Information Processing Systems 33 (2020): 8078-8088.
> >
> > [9] Gorishniy, Yury, et al. "Revisiting deep learning models for tabular data." Advances in Neural Information Processing Systems 34 (2021): 18932-18943.
> >
> > [10] Grinsztajn, Léo, Edouard Oyallon, and Gaël Varoquaux. "Why do tree-based models still outperform deep learning on typical tabular data?." Advances in Neural Information Processing Systems 35 (2022): 507-520.
> >
> > [11] McElfresh, Duncan, et al. "When Do Neural Nets Outperform Boosted Trees on Tabular Data?." arXiv preprint arXiv:2305.02997 (2023).
> >
> > [12] Kadra, Arlind, et al. "Well-tuned simple nets excel on tabular datasets." Advances in neural information processing systems 34 (2021): 23928-23941.
> >
> > [13] Chen, Tianqi, and Carlos Guestrin. "Xgboost: A scalable tree boosting system." Proceedings of the 22nd acm sigkdd international conference on knowledge discovery and data mining. 2016.
> >
> > [14] Pasad, Ankita, Ju-Chieh Chou, and Karen Livescu. "Layer-wise analysis of a self-supervised speech representation model." 2021 IEEE Automatic Speech Recognition and Understanding Workshop (ASRU). IEEE, 2021.
> >
> > [15] Liu, Shiwei, et al. "Sparsity May Cry: Let Us Fail (Current) Sparse Neural Networks Together!." arXiv preprint arXiv:2303.02141 (2023).
> >
> > [16] Chizat, Lenaic, Edouard Oyallon, and Francis Bach. "On lazy training in differentiable programming." Advances in neural information processing systems 32 (2019).

---

> > > ### Author Response · Authors · 2023-11-22
> > > **Gentle Reminder on Discussion Deadline**
> > >
> > > Dear Reviewer LdpC,
> > >
> > > Thank you again for your helpful review. Based on your suggestions, we spent much effort in experiments and writing to address your concerns. As the deadline for discussion nears, are there any additional concerns or questions you still have regarding our work? We are eager to clarify any remaining concerns or questions.

---

### Official Review · Reviewer_paCL · 2023-11-08

**Soundness:** 3 good
**Presentation:** 2 fair
**Contribution:** 2 fair
**Rating:** 3
**Confidence:** 3

**Summary:**

This paper proposes to prune the original dense network (supernetwork) to find an invariance-preserving subnetwork, called IUNet.
Toward this, the authors introduce the Invariance Learning Objective (ILO) that helps to learn representation invariant to
several data transformations and Proactive Initialization (PI).
The authors show that minimizing constrastive learning objective is equivalent to minimizing the distance between
representations of inputs under a set of invariant perturbations, thereby devising ILO.
Through several experiments on vision and tabular tasks, this paper validates the effectiveness of proposed IUNet and
conducted several empirical analysis.

**Strengths:**

It is novel to me that this paper proposes to obtain invariance-preserving network "via network pruning".
In reality, the experimental results on various tasks show the superior consistency measure, which is much higher than the baselines.
The theoretical insight between the invariant perturbations and the contrastive learning objective is also impressive.

**Weaknesses:**

Here are my main concerns.

1. The authors try to find the invariance-preserving network "via network pruning", but the proposed ILO could also be applied to just desne networks without network pruning. In this sense, there seems to be a lack of motivation as to why network pruning is necessary for obtaining an invariance-preserving network. Additionally, with magnitude-based one-shot pruning, a method simply based on parameter magnitude will be very sensitive to initialization. To alleviate this, more robust methods for initialization, such as SNIP and GraSP, have been proposed, but in this paper, there is no discussion or experiment on how sensitive to initialization.

2. It seems that many presentations of the paper should be revised. For example, in Section 3.2, the explanation for relations of parameters of supernetwork and subnetwork are quite confusing. Since $\theta_P^{(0)}$ and $\theta_M^{(T)}$ are parameter "vectors", hence the set inclusion operation is not suitable. Also, the expression $|\theta_P^{(0)}| < |\theta_M^{(T)}|$ does not imply that $\theta_P^{(0)}$ is more sparse than $\theta_M^{(T)}$, which does not give any information about the network capacity. Rather, the authors should express the sparsity or network capacity via $\ell_0$-norm (number of non-zeros, nnz). In the same sense, in Section 3.2.1, it seems difficult to say that the higher weight value of criterion (3) is a criterion of network capacity. Even if the parameter value is small, if it is non-zero, it will eventually be counted in the FLOPs calculation, so (for example) the number of non-zeros should be used rather than weight magnitude.

3. In the case of proactive initialization (PI), it is as if only the parameter size was simply adjusted, but looking at the results in Figure 3, the parameters were initialized small with $\kappa = 0.125$. It is natural that the result of $\kappa = 0.125$ has many weights with smaller values. There needs to be some mathematical evidence or experimental results for proactive initialization that can further demonstrate motivation and reasoning about it.

4. It is true that IUNet shows good performance in various experimental results, but it is difficult to interpret Figure 4 that the weights have any pattern. In this sense, I recommend that the authors consider a slightly larger size network parameter or describe more details about which parts are different between the first row and the second row in Figure 4.

**Questions:**

Please refer to the weaknesses.

---

> ### Author Response · Authors · 2023-11-21
> **Rebuttal Response Part 1**
>
> Thank you for your review! We provided additional experiments and clarifications to address your concerns. Please let us know if you have any remaining doubts!
>
> **[Motivation for Pruning]**
>
> Our goal is to improve the **network architecture’s inductive bias**.
>
> We agree weights play an important role in preserving invariance; however, the **network architecture’s inductive bias** is **independent of its weights** and refers to the inductive bias captured by **the structure of the network**. Improving the network architecture’s structure will improve its prior and optimization trajectory[1] **leading to better trained models**. In fact convolutional nets can be used as feature extractors **even under random initialization**[1,2,3]. Learning invariance by solely updating the weights (i.e. applying ILO *“without network pruning”*) will not achieve such properties, hence pruning is necessary.
>
> We empirically verified IUNet’s pruned network has **a better structure and inductive bias in our ablation studies, Table 4.** Our results show the pruned subnetwork achieves **substantially better** downstream accuracy than both the supernetwork trained normally and the supernetwork trained with our novelties, PIs and ILO. Hence, **IUNet indeed finds a pruned subnetwork with better inductive bias (or prior) than that of the supernetwork**.
>
> We will update our main text to emphasize the necessity of pruning clearer.
>
> **[On “SNIP” and “GRaSP”]**
>
> We agree that comparing IUNet with SNIP and GraSP can strengthen our claim that IUNet improves general pruning performance.
>
> To this end, **we compare IUNet against SNIP and GRaSP** on the SMC-Benchmark[4], which prunes the pretrained RoBERTa[5] on the Common Sense QA dataset[6]. Using the official Github repo, *https://github.com/VITA-Group/SMC-Bench/*, we reran SMC’s provided pruning algorithms OMP (after), OMP (before), and SNIP[7]. Additionally, we implemented and ran IUNet and GRaSP[8] on this benchmark. We report the accuracy under different sparsities.
>
> For a fair comparison, we leave out ILO from IUNet and adopt the same finetuning protocol and hyperparameters as SMC. We tried 2 different settings for kappa: [0.125, 0.0625]. Because SMC uses pretrained models: (1) IUNet scales the pretrained weights rather than random initialization by kappa, (2) IUNet reinitializes the pruned models weights to the fine-tuned model initialized from pretrained weights, not random initializations.
>
> Note, IUNet is a **post-tuning pruning algorithm** (similar to OMP (after) in the benchmark paper), where the model is first pretrained, then fine-tuned (either with maximum likelihood or ILO), then pruned, then sparsely fine-tuned to convergence. OMP(before), SNIP, and GRaSP are **pruning at initialization (PaI)**[4, 9] algorithms, where the model first pretrained, then pruned (iterating through the train set once) then sparsely fine-tuned to convergence.
>
>  ```
> Method\Sparsity   | 0.2   | 0.36  | 0.488 | 0.590 | 0.672
> —-----------------+-------+-------+-------+-------+-------
> OMP (Before)      | 26.54 | 58.89 | 25.88 | 19.57 | 19.57
> SNIP              | 18.67 | 20.48 | 18.84 | 19.00 | 19.16
> GRaSP             | 21.79 | 19.25 | 18.43 | 17.77 | 20.63
> —-----------------+-------+-------+-------+-------+-------
> OMP (After)       | 76.09 | 73.87 | 30.14 | 19.57 | 19.57
> —-----------------+-------+-------+-------+-------+-------
> IUNet* (k=0.125)  | 75.10 | 74.53 | 34.89 | 19.57 | 19.57
> IUNet* (k=0.0625) | 75.02 | 74.37 | 53.80 | 19.57 | 19.57
> *no-ILO
> ```
>
> The densely fine-tuned model reaches an accuracy of 75.76%.
>
> At low sparsities (sparsity= [0.2, 0.36]), **all post-tuning pruning algorithms (OMP(After) and IUNet) outperform PaI algorithms (OMP(Before), SNIP, GRaSP)**. At these sparsities, *OMP(After) and IUNet are within experimental uncertainty of each other*. This agrees with prior work[4, 9] which finds PaI is less effective than post-tuning algorithms.
>
> At middle sparsities (sparsity=0.488), **IUNet outperforms all baselines (OMP (after), OMP (before), SNIP, GRaSP),** because PIs alleviates the lazy learning phenomenon which is a key bottleneck in existing pruning algorithms[4]. This is a notable finding, showing **IUNet scales to larger sparsities than baselines.**
>
> At high sparsities (sparsity= [0.5, 0.672]), **all methods fail**, which agrees with SMC[4], suggesting at least 50% of weights are crucial to the transformer models on these tasks.
>
> **Hence, as shown in results above, IUNet generalizes to pre-trained models and outperforms both SNIP and GRaSP.**

---

> ### Author Response · Authors · 2023-11-21
> **Rebuttal Response Part 2**
>
> **[On Sensitivity to Initialization]**
>
> We ran all experiments 3 times from scratch under different random seeds for all methods. Hence, the trends we report between IUNet and OMP already consider OMP’s sensitivity to initialization.
>
> **[On Paper Presentation]**
>
> Thank you for pointing out this confusing notation. In Section 3.2, we will update our notation from $\theta$ to $\omega$, where $\omega$ denotes the set of parameters in each network: $|\omega_P|<<|\omega_M|$ and $\omega_P \subset \omega_M$.
>
> Your understanding of the l0-norm is correct, because the l0-norm of parameters where pruned parameters are zero is equivalent to the size of the set of remaining parameters. More formally:
>
> $$
> nnz(\theta_M) = \sum_{\theta_{Mi} \in \theta_{M}} 1[\theta_{Mi} \ne 0] = \sum_{\omega_{Pi} \in \omega_P} 1 = |\omega_P|
> $$
>
> **[On PIs]**
>
> PIs will scale the original weights by multiplying the Kaiming-initialized weights by a kappa scalar as stated in Section 3.2.2:
>
> $\theta_M^{(0)} = \kappa \theta_{M \dagger}^{(0)} $
>
> **We agree it is natural thus for the learned weights to also have smaller magnitudes!** This accomplishes our goal of filtering out unimportant weights, because most weights will have small magnitudes, while only important weights will have large magnitudes. In Figure 3, **we empirically verify** that the magnitude of important weights are the same (large) with and without PIs, yet the magnitude of most weights are small **only with PIs.**
>
> **Encouraging most weights to be small matters**, because the change in output logits is proportional to the magnitude of pruned weights. This relation between the magnitude of pruned weights and output logits can be seen in a simple 2 layer neural network with k-Lipschitz activation:
>
> Let the 2-layer supernetwork be: $f(x) = \sigma(\sigma(xW_{1})W_2), x\in R^d, W_1 \in R^{d\times d}, W_2 \in R^{d\times 1}$.
>
> Let the pruned subnetwork be: $g(x) = \sigma(\sum_{i\in P_2'}\sigma(\sum_{j,i\in P_1'}xW_{1,j,i})W_{2,i}))$ and pruned weight indices be $P_2 = argtopk(-|W_2|), P_1 = argtopk(-|W_1|)$ and unpruned weight indices be $P_2' = ${$ 1,...,d $}$ \setminus P_2, P_1' = ${$ 1,...,d $}$ ^2 \setminus P_1$.
>
> Then the difference in the pruned network’s and original network’s output logits is:
>
> $|f(x) - g(x)| \le k|\sigma(xW_{1})W_2 - \sum_{i\in P_2'}\sigma(\sum_{j,i\in P_1'}xW_{1,j,i})W_{2,i}|$
>
> $$|
> f(x) - g(x)| \le k C_1 + k^2 C_2
> $$
>
> where $C_1 = |\sum_{i \in P_2} \sigma(xW_{1})_i|$
>
> and $C_2 = \sum_{i\in P_2'} (W_{2,i}\sum_{j,i\in P_1}|xW_{1,j,i}|)$
>
> $|f(x) - g(x)| = \mathcal{O}(\sum_{i\in P_2}|W_{2,i}|+\sum_{j,k\in P_1}|W_{1,j,k}|)$
>
> **Figure 2abc presents empirical evidence of PIs efficacy**. As we decrease kappa, there is a **tradeoff** between performance of the unpruned network and performance improvement in the pruned network. Interestingly, **kappa has a sweet spot, where the pruned network’s performance grows higher than that of the Kaiming-initialized (kappa=1.0) supernetwork’s!** We highlight this is **a substantial finding among the network pruning literature**[10] and supports our hypothesis that through pruning, the subnetwork can improve the inductive bias of the supernetwork.
>
> **Table 4, ablation studies, presents empirical evidence of PIs efficacy**. If we train a model without PIs, performance is worse on computer vision datasets, as the pruning stage cannot differentiate important from unimportant weights. Tabular datasets show less consistent results since they are much more varied in size and feature type [14, 15, 16, 17]. In some tabular datasets, regularization becomes the bottleneck rather than model architecture [15, 16, 17]. Nonetheless, our ablation studies demonstrate **PIs will improve the downstream accuracy of IUNet**.
>
> **“Our [On “SNIP” and “GRaSP”] rebuttal section also presents empirical evidence of PIs efficacy.** By simply adjusting the initialization, **PIs scales transformers to larger sparsities than that of baselines**.
>
> We will update section 5.4 to make this empirical evidence more salient.

---

> ### Author Response · Authors · 2023-11-21
> **Rebuttal Response Part 3**
>
> **[On Figure 4]**
>
> We provide further discussion on the visualizations below and will update our text accordingly.
>
> In Figure 4, we visualize randomly-chosen filters from the first layer of each network. We randomly chose the filters for fairness, and picked the first layer because it is the easiest to interpret. The main conclusion from Figure 4 is: **IUNet bridges the gap between MLPs and CNNs and decision trees architecturally**. We make the following specific observations:
>
> **1) Locally Connected Regions:** For Figures 4efg, IUNet finds locally connected regions with high RGB values. Local connections help preserve translation and scale invariance as shown in Table 3 and verified by previous works[1, 2, 3, 11]. The presence of locally-connected regions visually confirms that IUNet can rediscover CNN architectures from MLPs.
>
> **2) Color Invariance:** In Figures 4eg (CIFAR10/SVHN), IUNet assigns high values to locally connected regions uniformly across all RGB channels, which improves color invariance. In Figure 4f (CIFAR100), this is not as consistent due to a more challenging learning objective, though we find it still improves color invariance over the MLP supernetwork.
>
> **3) Discovering Substructures:** Nearly all IUNet output neurons favor a clear pattern of input neurons (the exception being the top left filter of Figure 4f), whereas nearly no OMP output neurons exhibit any pattern. This suggests IUNet does a good job recovering invariance preserving substructures across the supernetwork’s first layer.
>
> **4) Axis-Aligned Tree Structures:** For Figure 4h, IUNet discovers subnetworks that focus on primarily 1 input feature for each output neuron. This behavior mirrors tree-based models where neurons are activated where single features reach some threshold [12, 13, 14]. Such network architecture follows existing work which found the axis-aligned inductive bias in tree-based models is critical to XGBoost’s performance over standard MLP setups [14].
>
> **[References]**
>
> [1] Ulyanov, Dmitry, Andrea Vedaldi, and Victor Lempitsky. "Deep image prior." Proceedings of the IEEE conference on computer vision and pattern recognition. 2018.
>
> [2] Cao, Yun-Hao, and Jianxin Wu. "A random cnn sees objects: One inductive bias of cnn and its applications." Proceedings Of The AAAI Conference On Artificial Intelligence. Vol. 36. No. 1. 2022.
>
> [3] Rosenfeld, Amir, and John K. Tsotsos. "Intriguing properties of randomly weighted networks: Generalizing while learning next to nothing." 2019 16th conference on computer and robot vision (CRV). IEEE, 2019.
>
> [4] Liu, Shiwei, et al. "Sparsity May Cry: Let Us Fail (Current) Sparse Neural Networks Together!." arXiv preprint arXiv:2303.02141 (2023).
>
> [5] Liu, Yinhan, et al. "Roberta: A robustly optimized bert pretraining approach." arXiv preprint arXiv:1907.11692 (2019).
>
> [6] Talmor, Alon, et al. "Commonsenseqa: A question answering challenge targeting commonsense knowledge." arXiv preprint arXiv:1811.00937 (2018).
>
> [7] Lee, Namhoon, Thalaiyasingam Ajanthan, and Philip HS Torr. "Snip: Single-shot network pruning based on connection sensitivity." arXiv preprint arXiv:1810.02340 (2018).
>
> [8] Wang, Chaoqi, Guodong Zhang, and Roger Grosse. "Picking winning tickets before training by preserving gradient flow." arXiv preprint arXiv:2002.07376 (2020).
>
> [9] Frankle, Jonathan, et al. "Pruning neural networks at initialization: Why are we missing the mark?." arXiv preprint arXiv:2009.08576 (2020).
>
> [10] Ma, Xiaolong, et al. "Sanity checks for lottery tickets: Does your winning ticket really win the jackpot?." Advances in Neural Information Processing Systems 34 (2021): 12749-12760.
>
> [11] Neyshabur, Behnam. "Towards learning convolutions from scratch." Advances in Neural Information Processing Systems 33 (2020): 8078-8088.
>
> [12] Marton, Sascha, et al. "GRANDE: Gradient-Based Decision Tree Ensembles." arXiv preprint arXiv:2309.17130 (2023).
>
> [13] Marton, Sascha, et al. "GradTree: Learning Axis-Aligned Decision Trees with Gradient Descent." NeurIPS 2023 Second Table Representation Learning Workshop. 2023.
>
> [14] Grinsztajn, Léo, Edouard Oyallon, and Gaël Varoquaux. "Why do tree-based models still outperform deep learning on typical tabular data?." Advances in Neural Information Processing Systems 35 (2022): 507-520.
>
> [15] Gorishniy, Yury, et al. "Revisiting deep learning models for tabular data." Advances in Neural Information Processing Systems 34 (2021): 18932-18943.
>
> [16] Kadra, Arlind, et al. "Well-tuned simple nets excel on tabular datasets." Advances in neural information processing systems 34 (2021): 23928-23941.
>
> [17] McElfresh, Duncan, et al. "When Do Neural Nets Outperform Boosted Trees on Tabular Data?." arXiv preprint arXiv:2305.02997 (2023).

---

> ### Comment · Reviewer_paCL · 2023-11-22
> **Response to Authors**
>
> I appreciate the authors for their kind rebuttals, but they only partially address my concerns.
>
> I understand why Proactive initialization helps the network pruning in theory (though the authors simply justify only for 2-layer networks) and practice. Also, the descriptions on Figure 4 helps me understand the points than before reading the rebuttals.
>
> However, I have still concern about "how and why pruning is required for learning invariance" (I think it is the most important part since it is the motivation of this paper). The authors say that inductive bias heavily depends on the architecture itself rather than on the model parameter, but this paper fully utilizes the pretrained models, in which one trains model at "good initialization".
>
> According to author's claim, I think that this paper should conduct the experiments without pretrained models, i.e., training from scratch with Invariance Learning Objective (ILO). Furthermore, in my view, ILO still seems far away from network pruning, and applying ILO to dense models (without pruning) is expected to better performance than pruned networks.
>
> Considering these points, I will maintain my score in the current shape.

---

> ### Author Response · Authors · 2023-11-22
> **Important Clarification**
>
> Thank you for your timely response! We want to make an important clarification.
>
> As stated in Section 3.2, our paper experiments **are** run on **randomly initialized networks**, not pretrained models.
>
> Specifically, in the paper, IUNet (1) **randomly initializes** the supernetwork with weights $\theta_{M^\dagger}^{(0)}$, (2) scales the random weights with PIs, $\theta_M^{(0)} = \kappa \theta_{M^\dagger}^{(0)}$, (3) trains the supernetwork **from scratch** using ILO to get $\theta_M^{(T)}$, (4) prunes the trained supernetwork into a subnetwork, (5) **randomly re-initializes** the pruned subnetwork as $\theta_{P}^{(0)}$ with Lottery Ticket Reinitialization (i.e. resetting weights to that of $\theta_{M^\dagger}^{(0)}$), and (6) finetunes the **randomly re-initialized** subnetwork **from scratch** using maximum likelihood to get $\theta_P^{(T)}$.
>
> Tables 1 and 2 compare the **randomly initialized** supernetwork and the **randomly initialized** pruned network (IUNet) both trained under the **same** maximum likelihood objective, where we find *"the inductive bias heavily depends on the architecture itself rather than on the model parameters"*. Table 4 shows the pruned networks (IUNet) outperforms applying ILO to dense models without pruning (IUNet (No-Prune)).
>
> We will update our notation and wording in Section 3.2 to make this clearer. Please let us know if you have any remaining questions. Thanks again!

---

> ### Comment · Reviewer_paCL · 2023-11-22
> **Response to Authors**
>
> That's my confusion, thanks for the correction to the authors. Also, I appreciate the authors for their response.

---

> > ### Author Response · Authors · 2023-11-22
> > **Thank you for your Response**
> >
> > We are glad our response has addressed your confusion! Please let us know if anything else we can further clarify!

---

### Meta-Review · Area_Chair_eiwD · 2023-12-06

**Metareview:**

The paper introduces a novel approach to achieving invariance-preserving networks through network pruning. Three reviewers have reviewed the paper and generally, they tended to agree that the paper demonstrates superior consistency measures across various tasks and offers valuable theoretical insights. The comprehensive exploration of invariance in neural network architectures and the proposal of a training objective for tabular data are notable strengths. However, concerns arise about the necessity and motivation for pruning in learning invariance. The weight initialization technique lacks novelty and contradicts claims about the dependence of inductive bias on network structure. Throughout the discussion among reviewers, they also all agree that the paper also suffers from a lack of originality, occasional errors, and unclear experimental details, raising doubts about the empirical validation of key claims.

While the paper presents promising aspects, the concerns regarding pruning's necessity and contradictory claims diminish its overall impact. These concerns makes the paper less convincing in its current form, necessitating further clarification and validation for a meaningful contribution to the field.

**Justification For Why Not Higher Score:**

all reviewers are negative and they are not convinced by the motivation of this paper.

**Justification For Why Not Lower Score:**

n/a

---

### Decision · Program_Chairs · 2024-01-16

Reject